# Action sequence learning, habits, and automaticity in obsessive-compulsive disorder

**Paula Banca[1,2]\*, Maria Herrojo Ruiz[3], Miguel Fernando Gonzalez-Zalba[4], Marjan Biria[1,2], Aleya A Marzuki[1,2], Thomas Piercy[5], Akeem Sule[5], Naomi A Fineberg[6,7], Trevor W Robbins[1,2]**

[1]Department of Psychology, University of Cambridge, Cambridge, United Kingdom; [2]Behavioural and Clinical Neuroscience Institute, University of Cambridge, Cambridge, United Kingdom; [3]Department of Psychology, Goldsmiths University of London, London, United Kingdom; [4]Quantum Motion, London, United Kingdom; [5]Department of Psychiatry, School of Clinical Medicine, University of Cambridge, Cambridge, United Kingdom; [6]Hertfordshire Partnership University NHS Foundation Trust, Welwyn Garden City, United Kingdom; [7]University of Hertfordshire, Hatfield, United Kingdom

**\*For correspondence:**
paula.banca@gmail.com

**Abstract** This study investigates the goal/habit imbalance theory of compulsion in obsessive-compulsive disorder (OCD), which postulates enhanced habit formation, increased automaticity, and impaired goal/habit arbitration. It directly tests these hypotheses using newly developed behavioral tasks. First, OCD patients and healthy participants were trained daily for a month using a smartphone app to perform chunked action sequences. Despite similar procedural learning and attainment of habitual performance (measured by an objective automaticity criterion) by both groups, OCD patients self-reported higher subjective habitual tendencies via a recently developed questionnaire. Subsequently, in a re-evaluation task assessing choices between established automatic and novel goal-directed actions, both groups were sensitive to re-evaluation based on monetary feedback. However, OCD patients, especially those with higher compulsive symptoms and habitual tendencies, showed a clear preference for trained/habitual sequences when choices were based on physical effort, possibly due to their higher attributed intrinsic value. These patients also used the habit-training app more extensively and reported symptom relief post-study. The tendency to attribute higher intrinsic value to familiar actions may be a potential mechanism leading to compulsions and an important addition to the goal/habit imbalance hypothesis in OCD. We also highlight the potential of smartphone app training as a habit reversal therapeutic tool.

## eLife assessment

This study provides **solid** evidence for differences in habit-learning in obsessive-compulsive disorder versus controls. Contrary to previous studies that employed a single laboratory session to study habit-learning, here a smartphone app delivered motor-sequence tasks daily for a month. These results have **important** implications for our understanding of goal-directed versus habit learning in obsessive-compulsive disorder.

## Introduction

Considerable evidence has supported the concept of imbalanced cortico-striatal pathways mediating compulsive behavior in obsessive-compulsive disorder (OCD). This imbalance has been suggested to reflect a weaker goal-directed control and an excessive habitual control (*Gillan et al., 2016*).

Dysfunctional goal-directed control in OCD has been strongly supported both behaviorally (*Gillan et al., 2011*; *Vaghi et al., 2019*) and from a neurobiological perspective (*Gillan et al., 2015a*). However, until now, enhanced (and potentially maladaptive) habit formation has largely been inferred by the absence of goal-directed control, although recent studies show increased self-reported habitual tendencies in OCD, as measured by the Self-Report Habit Index Scale (*Ferreira et al., 2017*). Problems with this 'zero-sum' hypothesis (*Robbins and Costa, 2017*) (i.e. diminished goal-directed control *thus* enhanced habitual control) have been reiterated by recent findings linking stimulus-response strength (*Zwosta et al., 2018*) and goal devaluation (*Gillan et al., 2015b*) exclusively to a dysfunctional goal system. There is thus a need to focus specifically on the habit component of the associative dual-process (i.e. goal/habit) model of behavior and test more directly the hypothesis of enhanced habit formation in OCD.

We recently proposed that extensive training of sequential actions could be a means for rapidly engaging the 'habit system' in a laboratory setting (*Robbins et al., 2019*). The idea is that, in action sequences (like those seen in skilled routines), extensive training helps integrate separate motor actions into a coordinated and unified sequence, or 'chunk' (*Graybiel, 1998*; *Sakai et al., 2003*). Through consistent practice, the selection and execution of these component actions become more streamlined, stereotypical, and cognitively effortless. They are performed with minimal variation, achieving high efficiency. Moreover, there is now robust evidence that for highly trained sequences, actions are represented in parallel according to their serial order before execution (*Kornysheva et al., 2019*). Such features relate to the concept of *automaticity*, which captures many of the shared elements between habits and skills (*Ashby et al., 2010*). At a neural level, automaticity is associated with a shift in control from the anterior/associative (goal-directed) to the posterior/sensorimotor (habitual) striatal regions (*Ashby et al., 2010*; *Graybiel and Grafton, 2015*; *Kupferschmidt et al., 2017*), accompanied by a disengagement of cognitive control hubs in frontal and cingulate cortices (*Bassett et al., 2015*). In fact, within the skill learning literature, this progressive shift to posterior striatum has been linked to the gradually attained asymptotic performance of the skill (*Bassett et al., 2015*; *Doyon et al., 2018*; *Doyon et al., 2015*; *Lehéricy et al., 2005*). Hence, chunked action sequences provide an opportunity to target the brain's goal-habit transition and study the relationships between automaticity, skills, and habits (*Dezfouli et al., 2014*; *Graybiel and Grafton, 2015*; *Robbins and Costa, 2017*). This approach is relevant for OCD research as it mimics the sequences of motor events and routines observed in typical compulsions, often performed in a "just right" manner (*Hellriegel et al., 2017*), akin to skill learning. Chunked action sequences also enable investigation of the relationships between hypothesized procedural learning deficits in OCD (*Rauch et al., 1997*) and automaticity.

Following this reasoning, we developed a smartphone *Motor Sequencing App* with attractive sensory features in a game-like setting, to investigate automaticity and measure habit/skill formation within a naturalistic setting (at home). This task, akin to a piano-based app, allows subjects to learn and practice two sequences of finger movements. It was tailored to emphasize the positive aspects of habits, as advocated by *Watson et al., 2022*, and it satisfies central criteria that define habits proposed by *Balleine and Dezfouli, 2019*: swift execution, invariant response topography, and action chunking. We also aimed to investigate within the same experiment three facets of automaticity which, according to *Haith and Krakauer, 2018*, have rarely been measured together: habit, skill, and cognitive load. Although there is no consensus on how exactly skills and habits interact (*Robbins and Costa, 2017*), it is generally agreed that both lead to automaticity with sufficient practice (*Graybiel and Grafton, 2015*) and that the autonomous nature of habits and the fluid proficiency of skills engage the same sensorimotor cortical-striatal 'loops' (the so-called 'habit circuitry') (*Ashby et al., 2010*; *Graybiel and Grafton, 2015*). By focusing more on the *automaticity* of the response per se (as reflected by the speed and stereotypy of overtrained movement sequences), rather than on the *autonomous* nature of the behavior (an action that continues after a state change, e.g. devaluation of the goal), we do not solely rely on the devaluation criterion used in previous studies of compulsive behavior. This is important because outcome devaluation insensitivity as a test of habit in humans is controversial (*Watson et al., 2022*) and may indeed be a more sensitive indicator of failures of goal-directed control rather than of habitual control per se (*Balleine and Dezfouli, 2019*; *Robbins et al., 2019*; *Robbins and Costa, 2017*).

While designing our app, we additionally considered previous research emphasizing training frequency, context stability, and reward contingencies as important features for enhancing habit

strength (*Wood and Rünger, 2016*). To ensure effective consolidation required for habit/skill retention to occur, we implemented a 1-month training period. This aligns with studies showing that practice alone is insufficient for habit development as it also requires off-line consolidation over longer periods of time and sleep (*Nusbaum et al., 2018*; *Walker et al., 2003*). Finally, given the influence of reinforcer predictability on habit acquisition speed (*Bouton, 2021*), we employed two different reinforcement schedules (reward scores: continuous versus variable [probabilistic]) to assess their impact on habit formation among healthy volunteers (HV) and patients with OCD.

## Outline

In this article, we applied, for the first time, app-based behavioral training (experiment 1) to a sample of patients with OCD. We compared 32 patients and 33 healthy participants, matched for age, gender, IQ, and years of education in measures of motivation and app engagement (see Materials and methods for participants' demographics and clinical characteristics). We also assessed to what extent performing such repetitive actions in 1 month impacted OCD symptomatology. In an *initial phase* (30 days), two action sequences were trained daily to produce habits/automatic actions (experiment 1). We collected data online continuously to monitor engagement and performance in real time. This approach ensured we acquired sufficient data for subsequent analysis of procedural learning and automaticity development.

In a *second phase,* we administered two follow-up behavioral tasks (experiments 2 and 3) addressing two important questions relevant to the habit theory of OCD. The first research question investigated whether repeated performance of motor sequences could develop implicit rewarding properties, hence gaining value, potentially leading to compulsive-like behaviors (experiment 2: explicit preference task, conducted without feedback). The hypothesis postulates that the repeated performance, initially driven by the goal of proficiency, may eventually become motivated by its own implicit reward, tied to proprioceptive and kinesthetic feedback (e.g. offering anxiety relief alongside skillful execution). The second question explored whether manipulations of extrinsic feedback, based on monetary reward or on the physical effort required (by varying the length of the sequence) affected choice for the familiar trained action sequence (experiment 3: re-evaluation task, conducted with feedback).

Finally, we administered a comprehensive set of self-reported clinical questionnaires, including a recently developed questionnaire (*Ersche et al., 2017*) on habit-related aspects. This aimed to investigate: (1) if OCD patients report more habits; (2) whether stronger subjective habitual tendencies predict enhanced procedural learning, automaticity development, and an (in)ability to adjust to changing circumstances; and (3) if app-based habit reversal therapy yields therapeutic benefits or has any subjective sequelae in OCD.

## Hypothesis

Anticipating implicit learning issues in OCD (*Deckersbach et al., 2002*; *Kathmann et al., 2005*; *Rauch et al., 1997*) and fine-motor difficulties (*Bloch et al., 2011*), we expected poorer procedural learning in patients compared to HV. However, once learned, we predicted OCD patients would reach automaticity faster, possibly due to a stronger tendency to form habitual/automatic actions (*Gillan et al., 2016*; *Gillan et al., 2014*). We also hypothesized differences in the learning rate and automaticity development between the two action sequences based on their associated (1) reward schedule (continuous versus variable), with faster automaticity in the continuous reward sequence, as suggested by past research (*Bouton, 2021*); and (2) sign of changes in reward scores, expecting enhanced performance improvements following a decrease in scores, particularly pronounced in OCD patients due to heightened sensitivity to negative feedback (*Apergis-Schoute et al., 2024*; *Becker et al., 2014*; *Kanen et al., 2019*). Additionally, we predicted that OCD patients would generally display stronger habits and assign greater intrinsic value to the familiar app sequences, evidenced by a marked preference for executing them even when presented with a simpler alternative sequences. Finally, we expected patients to show a greater tendency to perform the familiar/trained sequences, even though its extrinsic relative value was reduced and new, more valuable, options became available.

## Results

### Self-reported habit tendencies

Participants completed self-reported questionnaires measuring various psychological constructs (see Materials and methods). Highly relevant for the current topic is the Creature of Habit Scale (COHS) (*Ersche et al., 2017*), recently developed to measure individual differences in *routine* behavior and *automatic* responses in everyday life. As compared to healthy controls, OCD patients reported significantly higher habitual tendencies in both the routine ($t = -2.79$, $p = 0.01$; HV: $\overline{COHSroutine} = 48.4$, $\sigma = 9$; OCD: $\overline{COHSroutine} = 55.7$, $\sigma = 11$) and the automaticity ($t = -3.15$, $p < 0.001$; HV: $\overline{COHSautomaticity} = 26.3$, $\sigma = 8$; OCD: $\overline{COHSautomaticity} = 32.9$, $\sigma = 9$) subscales.

### Phase A: Experiment 1

#### Motor sequence acquisition using the app

The task was a self-instructed and self-paced smartphone application (app) downloaded to participants' iPhones. It consisted of a motor practice program that participants committed to pursue daily, for a period of 1 month. An exhaustive description of the method has been previously published (*Banca et al., 2019*) but a succinct description can be found below, in *Figure 1* and in *Video 1*.

The training consisted of practicing two sequences of finger movements, composed of chords (two or three simultaneous finger presses) and single presses (one finger only). Each sequence comprised six moves and was performed using four fingers of the dominant hand (index, middle, ring, and little finger). Participants received feedback on each sequence performance (trial). Successful trials (to which we later refer as sequence trial number [*n*]) were followed by a positive ring tone and a positive visual effect (rewarding stars) and the unsuccessful ones by a negative ring tone and a negative visual effect (red lines on the screen). Every time a mistake occurred (irrespective of which move in the sequence it occurred), participants were prompted to restart the trial. Instructions were to respond swiftly and accurately. Participants were required to keep their fingers very close to the keys to minimize movement amplitude variation and to facilitate fast performance. To promote sequence learning and memorization, we implemented three progressively challenging practice levels. Initially (first three practice sessions), subjects responded to visual and auditory cues, following lighted keys associated with musical notes (level 1). As practice advanced, to enable motor independence and automaticity, these external cues were gradually removed: level 2 included only auditory cues (practices 4 and 5), and level 3 had no cues (remaining practices). Successful performance at each difficulty level resulted in progression to the next one. Unsuccessful performance led to reverting to the prior stage.

Each sequence, identified by a specific abstract image, was associated with a particular reward schedule. Points were calculated as a function of the time taken to complete a sequence trial. Accordingly, performance time was the instructed task-related dimension (i.e. associated with reward). In the *continuous reward schedule,* points were received for every successful trial whereas in the *variable reward schedule,* points were shown only on 37% of the trials. The rationale for having two distinct reward schedules was to assess their possible dissociable effect on the participants' development of automatic actions. For each rewarded trial, participants could see their achieved points on the trial. To increase motivation, the total points achieved on each training session (i.e. practice) were also shown, so participants could see how well they improved across practice and days. The permanent accessibility of the app (given that most people carry their mobile phones everywhere) facilitated training frequency and enabled context stability.

#### Practice schedule

All participants were presented with a calendar schedule and were asked to practice both sequences daily. They were instructed to practice as many times as they wished, whenever they wanted during the day and with the sequence order they would prefer. However, a minimum of two practices (*P*) per sequence was required every day; each practice comprised 20 successful

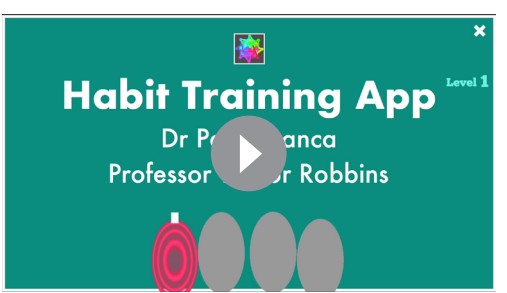

**Video 1.** Visual demonstration of the Motor Sequencing App for a better understanding of the task. https://elifesciences.org/articles/87346/figures#video1

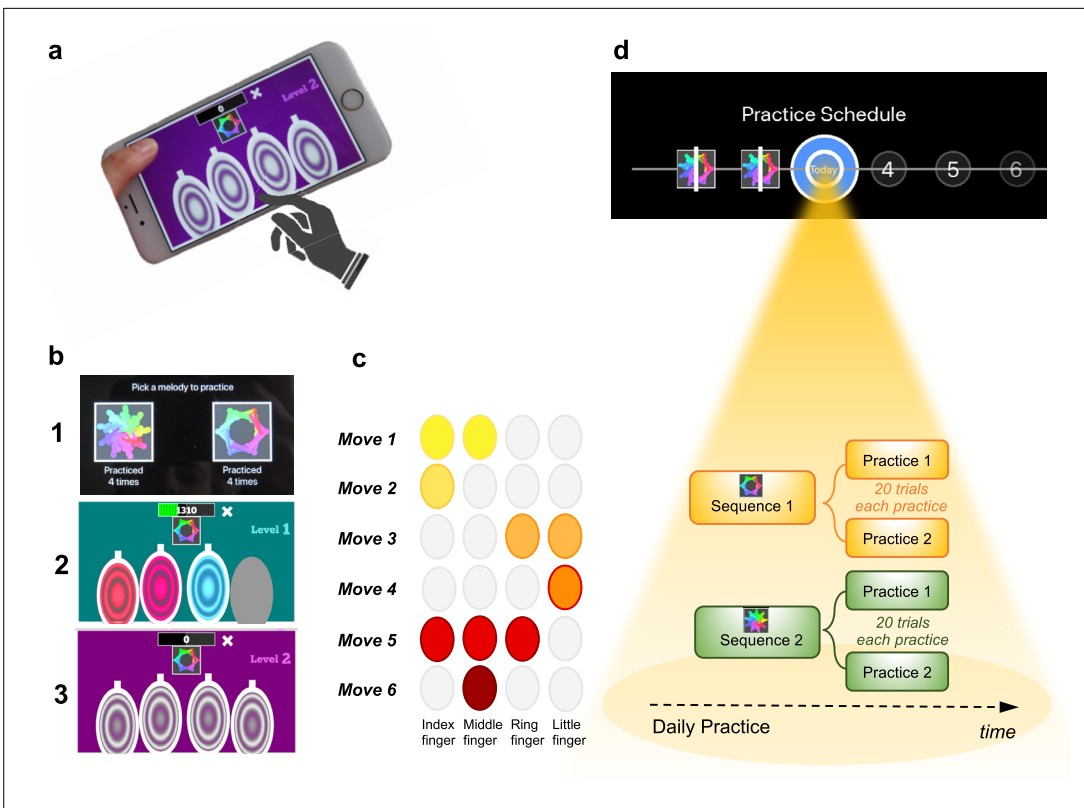

**Figure 1.** Motor Sequencing App. (**a**) A trial starts with a static image depicting the abstract picture that identifies the sequence to be performed (or 'played') as well as the four keys that will need to be tapped. Participants use their dominant hand to play the required keys: excluding the thumb, the leftmost finger corresponds to the first circle and the rightmost finger corresponds to the last circle. (**b**) Screenshot examples of the task design: (1) sequence selection panel, each sequence is identified by an abstract picture; (2) panel exemplifying visual cues that initially guide the sequence learning; (3) panel exemplifying the removal of the visual cues, when sequence learning is only guided by auditory cues. (**c**) Example of a sequence performed with the right hand: 6-moves in length, each move can comprise multiple finger presses (2 or 3 simultaneous) or a single finger press. Each sequence comprises 3 single press moves, 2 two-finger moves, and 1 three-finger move. (**d**) Short description of the daily practice schedule. Each day, participants are required to play *a minimum* of two practices per sequence. Each practice comprised 20 successful trials. Participants could play more if they wished and the order of the training practices was self-determined.

sequence trials. Participants had to make up for missed training by completing both the current day's session and the previous day's if they skipped a day. If they missed training for over 2 days, the researcher gauged their motivation and incentivized their commitment. Participants were excluded if they missed training for more than 5 consecutive days.

At least 30 days of training was required, and all data were anonymously collected in real time, through an online server. On the 21st day of practice, the rewards were removed (extinction) to ensure that the action sequences were more dependent on proprioceptive and kinesthetic, rather than on external, feedback. Analysis of the reward removal (extinction) is presented in Appendix 1 and *Appendix 1—figure 3*. Other additional task components and analysis are also included in Appendix 1 and *Appendix 1—figures 1 and 2*.

## Training engagement

Participants reliably committed to their regular training schedule, practicing consistently both sequences every day. Unexpectedly, OCD patients completed significantly more practices as compared with HV ($p = 0.005$) (*Figure 2a*). Descriptive statistics are as follows (values provided as median number of practices and interquartile range): HV: $\tilde{P} = 122$, *IQR* = 7; OCD: $\tilde{P} = 130$, *IQR* = 14. When visually inspecting the daily training pattern, we observed that HV tended to practice earlier

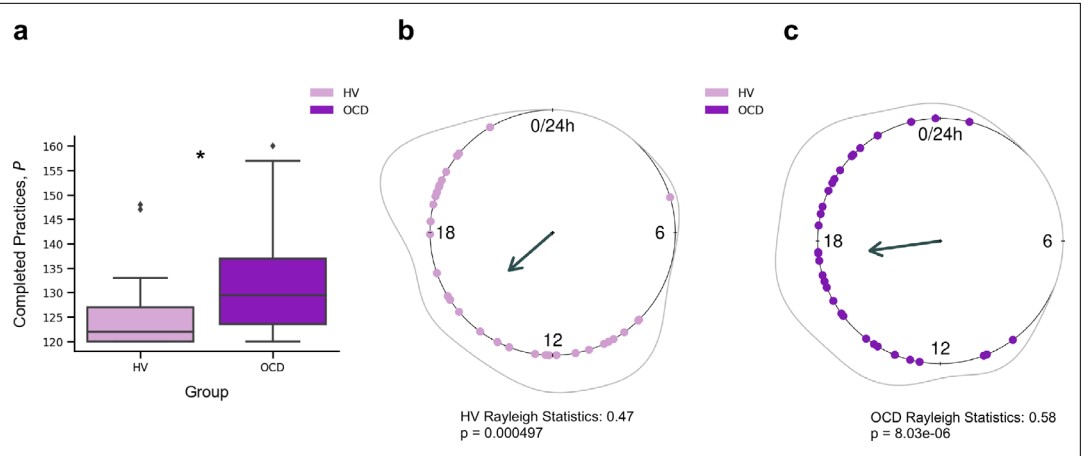

**Figure 2.** Training engagement. (**a**) Whole training overview. Obsessive-compulsive disorder (OCD, N = 32) patients engaged in significantly more training sessions than healthy volunteers (HV, N = 33) (*$p$ = 0.005). The minimum required practices ($P$) were 120$P$. (**b**) Daily training pattern for HV (N = 33) and (**c**) daily training pattern for OCD (N = 32). Single dots on the unit circle denote the preferred practice times of individual participants within 0–24 hr, obtained from the mean resultant vector of individual practice hours data (Rayleigh statistics). Group-level statistics were conducted in each group separately using the Rayleigh test to assess the uniformity of a circular distribution of points. The graphic displays the length of the mean resultant vector in each distribution, and the associated p-value. Regarding between-group statistical analysis, see main text.

than OCD. Circular statistics within each group demonstrated that HV practiced preferentially at a peak time of ~15:00 (mean resultant length 0.47, $p$ = 0.000497, Rayleigh test for the uniformity of a circular distribution of points; *Figure 2b*). For OCD participants, the preferred practice time had a mean direction at ~18:00 (mean resultant length 0.58, $p$ = 8.03 × 10$^{-6}$, Rayleigh test; *Figure 2c*). There were, however, no significant differences between both samples ($p$ = 0.19, Watson's $U^2$ test).

## Learning

Learning was evaluated by the decrement in sequence duration throughout training. To follow the nomenclature of the motor control literature, we refer to sequence duration as movement time ($MT$, in s), which is defined as

$$MT = t_6 - t_1,\qquad(1)$$

where $t_6$ and $t_1$ are the time of the last (6th) and first key presses, respectively.

For each participant and sequence reward type (continuous and variable), we measured $MT$ of a successful trial, as a function of the sequence trial number, $n$, across the whole training. Across trials, $MT$ decreased exponentially (*Figure 3a*). The decrease in $MT$ has been widely used to quantify learning in previous research (*Crossman, 1959*). A single exponential is viewed as the most statistically robust function to model such decrease (*Heathcote et al., 2000*). Accordingly, each participant's learning profile was modeled as follows:

$$MT(n) = MT_0 + MT_L \exp\left(-\frac{n}{n_r}\right),\qquad(2)$$

where $n_r$ is the *learning rate* (measured in number of trials), which governs the rate of exponential decay. Parameter $MT_0$ is the movement time at *asymptote* (at the end of the training). Last, $MT_L$ is the speed-up achieved over the course of the training (referred to as *amount of learning*) (*Figure 3a*). The larger the value of $MT_L$, the bigger the decline in the movement time and thus the larger the improvement in motor learning.

The individual fitting approach we used has the advantage of handling the different number of trials executed by each participant by modeling their behavior to a consolidated maximum value of $n$, $n_{max}$ = 1200. We used a moving average of 20 trials to mitigate any effect of outlier trials. This analysis was conducted separately for continuous and variable reward schedules.

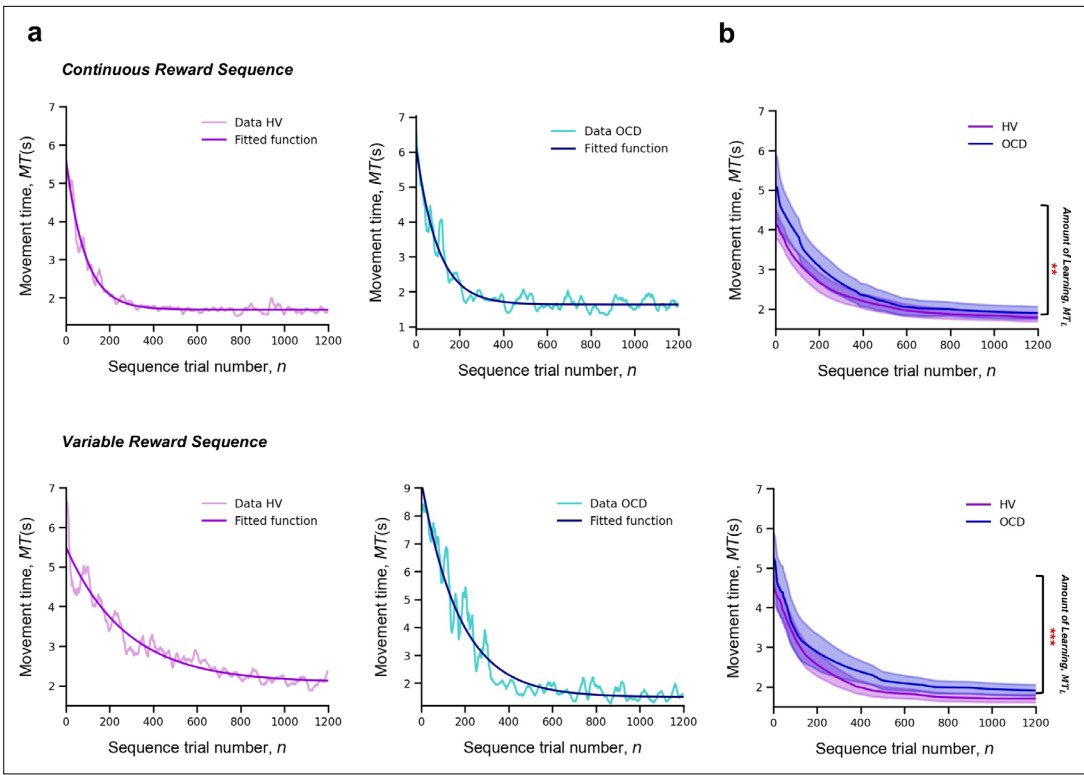

**Figure 3.** Learning. Upper panel: Model fitting procedure conducted for the continuous reward sequence. Lower panel: Model fitting procedure conducted for the variable reward sequence. (**a**) Individual plots exemplifying the time-course of $MT$ (in s) as training progresses (lighter color) as well as the exponential decay fit modeling the learning profile of a single participant (darker color). Left panels depict data in an healthy volunteers (HV) individual, right panels display data in a patient with obsessive-compulsive disorder (OCD). (**b**) Group comparison resulting from all individual exponential decays modeling the learning profile of each participant. A significant group difference was observed on the amount of learning, $MT_L$, in both reward schedule conditions (continuous: $p$ = 0.009; variable: $p$ < 0.001). Solid lines: median; transparent regions: median ± 1.57 × interquartile range/sqrt($n$); purple: HV (N = 33); blue: patients with OCD (N = 32).

To statistically assess between-group differences in learning behavior, we pooled the individual model parameters ($MT_L$, $n_r$ and $MT_0$), and conducted a Kruskal-Wallis $H$ test to assess the effect of group (HV and OCD), reward type (continuous and variable), and their interaction on each parameter (*Figure 3b*).

There was a significant effect of group on the *amount of learning* ($MT_L$) parameter, $H$ = 16.5, $p$ < 0.001, but no reward ($p$ = 0.06) or interaction effects ($p$ = 0.34) (*Figure 3c*). Descriptive statistics are as follows (values provided as median and interquartile range): HV: $\tilde{MT}_L$ = 3.1 s, $IQR$ = 1.2 s and OCD: $\tilde{MT}_L$ = 3.9 s, $IQR$ = 2.3 s for the continuous reward sequence; HV: $\tilde{MT}_L$ = 2.3 s, $IQR$ = 1.2 s and OCD: $\tilde{MT}_L$ = 3.6 s, $IQR$ = 2.5 s for the variable reward sequence.

Regarding the *learning rate* ($n_r$) parameter, we found no significant main effects of group ($p$ = 0.79), reward ($p$ = 0.47), or interaction effects ($p$ = 0.46). Descriptive statistics: sequence trials needed to asymptote HV: $\tilde{n}_r$ = 176, $IQR$ = 99 and OCD: $\tilde{n}_r$ = 200, $IQR$ = 114 for the continuous reward sequence; HV: $\tilde{n}_r$ = 182, $IQR$ = 123 and OCD: $\tilde{n}_r$ = 162, $IQR$ = 141 for the variable reward sequence. These non-significant effects on the learning rate were further assessed with Bayes factors ($BF$) for factorial designs (see Materials and methods). This approach estimates the ratio between the full model, including main and interaction effects, and a restricted model that excludes a specific effect. The evidence for the lack of main effect of group was associated with a $BF$ of 0.38, which is anecdotal evidence. We additionally obtained moderate evidence supporting the absence of a main effect of reward or a reward × group interaction ($BF$ = 0.16 and 0.17, respectively).

In analyzing the asymptote ($MT_0$) parameter, we found no significant main or interaction effects (group effect: $p$ = 0.17; reward effect: $p$ = 0.65 and interaction effect: $p$ = 0.64). Descriptive statistics

are as follows: HV: $\tilde{MT}_0$ = 1.7 s, IQR = 0.4 s and OCD: $\tilde{MT}_0$ = 1.8 s, IQR = 0.5 s for the continuous reward sequence; HV: $\tilde{MT}_0$ = 1.8 s, IQR = 0.5 s and OCD: $\tilde{MT}_0$ = 1.8 s, IQR = 0.5 s for the variable reward sequence. BF analysis indicated anecdotal evidence against a main group effect (BF = 0.53). Meanwhile, there was moderate evidence suggesting neither reward nor reward × interaction factors significantly influenced performance time (BF = 0.12 and 0.17, respectively).

The results indicate that OCD patients do not exhibit learning deficits. While they initially performed action sequences slower than the HV group, their learning rates ultimately matched those of HV. Both groups showed comparable movement durations at the asymptote. This suggests that, though OCD patients began at a lower baseline level of performance, they enhanced their motor learning to a degree that reached the same asymptotic performance as the controls.

## Automaticity

To assess automaticity, the ability to perform actions with low-level cognitive engagement, we examined the decline over time in the consistency of inter-keystroke interval (IKI) patterns trial to trial. We mathematically defined IKI consistency as the sum of the absolute value of the time lapses between finger presses from one sequence to the previous one.

$$C = \sum_{k=1}^{5} \left| t_{k,n+1} - t_{k,n} \right|, \tag{3}$$

where $n$ is the sequence trial number and $k$ is the inter-keystroke response interval (*Figure 4a*). In other words, $C$ quantifies how consistent/reproducible the press pattern is from trial to trial. The assumption here is that the more reproducible the sequences are *over time*, the more automatic the person's motor performance becomes.

For each participant and sequence reward type (continuous and variable), automaticity was assessed based on the decrement in $C$, as a function of $n$, across the entire training period. Since $C$ decreased in an exponential fashion, we fitted the $C$ data with an exponential decay function (following the same reasoning and procedure as *MT*) to model the automaticity profile of each participant,

$$C(n) = C_0 + C_L \exp\left(-\frac{n}{n_C}\right), \tag{4}$$

where $n_C$ is the *automaticity rate* (measured in number of trials), $C_0$ is the sequence consistency at *asymptote* (by the end of the training), and $C_L$ is the change in automaticity over the course of the training (which we refer to as *amount of automation gain*). The model fitting procedure was conducted separately for continuous and variable reward schedules.

A Kruskal-Wallis *H* test was then conducted to assess the effect of group (OCD and HV) and reward type (continuous and variable) on each parameter resulting from the individual exponential fits ($C_L$, $n_C$ and $C_0$).

There was a significant effect of group on the *amount of automation gain* ($C_L$) parameter: $H$ = 11.1, $p < 0.001$ but no reward ($p = 0.12$) or interaction effects ($p = 0.5$) (*Figure 4b*). Descriptive statistics are as follows: HV: $\tilde{C}_L$ = 1.4 s, IQR = 0.7 s and OCD: $\tilde{C}_L$ = 1.9 s, IQR = 1.0 s for the continuous reward sequence; HV: $\tilde{C}_L$ = 1.1 s, IQR = 0.8 s and OCD: $\tilde{C}_L$ = 1.5 s, IQR = 1.1 s for the variable reward sequence.

There was also a significant group effect on the *automaticity rate* ($n_C$) parameter: $H$ = 4.61, $p < 0.03$ but no reward ($p = 0.42$) or interaction ($p = 0.12$) effects. Descriptive statistics: sequence trials needed to asymptote HV: $\tilde{n}_C$ = 142, IQR = 122 and OCD: $\tilde{n}_C$ = 198, IQR = 162 for the continuous reward sequence; HV: $\tilde{n}_C$ = 161, IQR = 104 and OCD: $\tilde{n}_C$ = 191, IQR = 138 for the variable reward sequence.

At *asymptote* ($C_0$), no group ($p = 0.1$), reward ($p = 0.9$), or interaction ($p = 0.45$) effects were found. We found anecdotal evidence against a main group effect (BF = 0.65). In addition, there was moderate evidence in favor of no main effects of reward or interaction (BF = 0.12 and 0.18, respectively).

Of note is the median consistency in consecutive sequences achieved at asymptote: HV: $\tilde{C}_0$ = 287 ms, IQR = 127 ms, OCD: $\tilde{C}_0$ 301 ms, IQR = 186 ms for the continuous reward sequence and HV: $\tilde{C}_0$ = 288 ms, IQR = 110 ms, OCD: $\tilde{C}_0$ = 300 ms, IQR = 114 ms for the variable reward sequence. These values of the $C$ at asymptote are generally shorter than the normal reaction time for motor performance (*Kosinski, 2008*), reinforcing the idea that automaticity was reached by the end of the training.

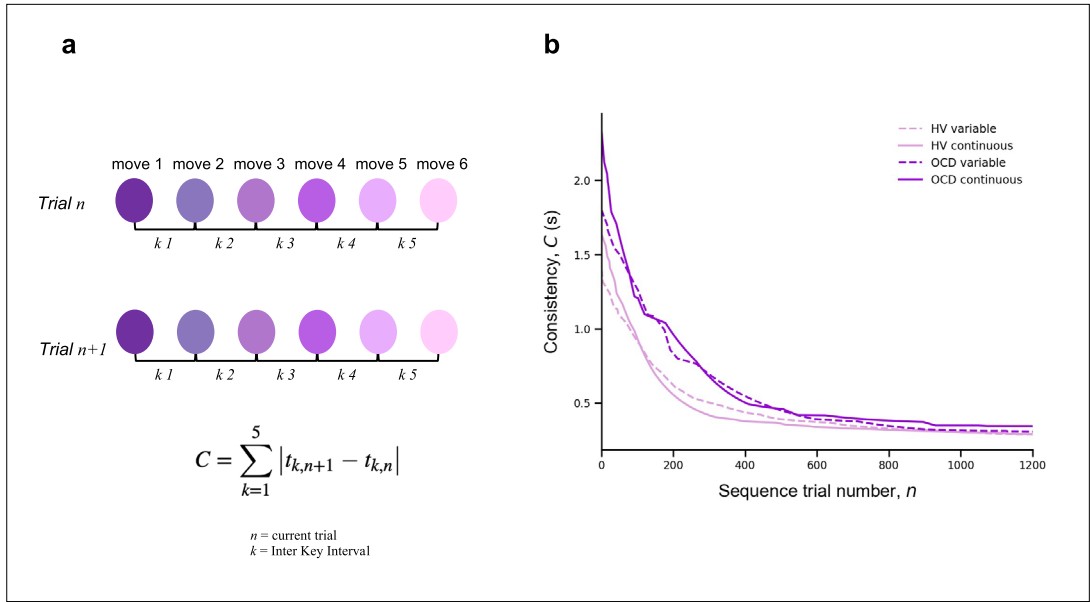

**Figure 4.** Automaticity. (**a**) We mathematically defined trial-to-trial inter-keystroke-interval consistency (IKI consistency), denoted as *C* (in s), as the sum of the absolute values of the time lapses between finger presses across consecutive sequences. The variable *n* represents the sequence trial and *k* denotes the IKI. We evaluated automaticity by analyzing the decline in *C* over time, as it approached asymptotic levels. (**b**) Group comparison resulting from all individual exponential decays modeling the automaticity profile (drop in *C*) of each participant. A significant group effect was found on the amount of automaticity gain, $C_L$ (Kruskal-Wallis $H = 11.1$, $p < 0.001$) and on the automaticity constant, $n_C$ (Kruskal-Wallis $H = 4.61$, $p < 0.03$). Solid and dashed lines are median values (*M*). Light purple: healthy volunteers (HV); dark purple: patients with obsessive-compulsive-disorder (OCD); solid lines: continuous reward condition; dashed lines: variable reward condition. Sample size (N): HV = 33, OCD = 32.

In conclusion, compared to HV, patients took significantly longer to achieve a similar level of automaticity in both reward schedules. They began at a slower pace, exhibited more variability, and progressed to automaticity at a slower rate.

## Sensitivity of sequence duration to reward

Our next goal was to investigate the sensitivity of performance improvements over time in our participant groups to changes in scores, whether they increased or decreased. To do this, we quantified the trial-by-trial behavioral changes in response to a decrement or increase in reward from the previous trial using the sequence duration (in ms), labeled as *MT* (movement time). Note that in our experimental design, *MT* was negatively correlated with the scores received. Following *Pekny et al., 2015*, we represented the change from trial *n* to *n*+1 in *MT* simply as:

$$\Delta MT^{(n+1)} = MT^{(n+1)} - MT^{(n)} \tag{5}$$

Reward (*R*) change at trial *n* was computed as:

$$\Delta R^{(n)} = R^{(n)} - R^{(n-1)} \tag{6}$$

We next aimed to analyze separately Δ*MT* values that followed an increase in reward from trial *n −* 1 to *n*, Δ*R+*, denoting a positive sign in Δ*R*; and those that followed a drop in reward, Δ*R−*, indicating a negative sign in Δ*R*. An issue arises with poor performance trials (those with a slower duration, or a large $MT^{(n)}$). These could inherently result in a systematic link between Δ*R−* and smaller (negative) $\Delta MT^{(n+1)}$ values due to the statistical effect known as 'regression to the mean'. Essentially, a trial that is poorly performed, marked by a large $MT^{(n)}$, is likely to be followed by a smaller $MT^{(n+1)}$ just because extreme values tend to be followed by values closer to the mean. As training progresses and *MT* reduces overall, the potential for significant changes relative to reward increments or decrements may

diminish. To account for and counteract this statistical artifact, we normalized the $\Delta MT^{(n+1)}$ index using the baseline $MT^{(n)}$:

$$\Delta MT^{(n+1)} = (MT^{(n+1)} - MT^{(n)})/MT^{(n)} \qquad (7)$$

We used this normalized measure of $\Delta MT^{(n+1)}$ (adimensional) for further analyses. It reflects the behavioral change from trial $n$ to $n+1$ relative to the baseline performance on trial $n$. Following *Pekny et al., 2015*, we estimated for each participant the conditional probability distributions $p(\Delta T|\Delta R+)$ and $p(\Delta T|\Delta R-)$ (where $T$ denotes a behavioral measure, $MT$ in this section or $IKI$ consistency in the next section) by fitting a Gaussian distribution to the histogram of each data sample (*Appendix 1—figure 5*). The standard deviation ($\sigma$) and the center $\mu$ of the resulting distributions were used for statistical analyses (*Appendix 1—figure 5*). Similar analyses were carried out on a normalized version of index $C$ (*Equation 3*), which already reflected changes between consecutive trials. See next section.

As a general result, we expected that healthy participants would introduce larger behavioral changes (more pronounced reduction in $MT$, more negative $\Delta MT$) following a decrease in scores, as shown previously (*Chen et al., 2017*; *van Mastrigt et al., 2020*). Accordingly, we predicted that the $p(\Delta T|\Delta R-)$ distribution would be centered at more negative values than $p(\Delta T|\Delta R+)$, corresponding to greater speeding following negative reward changes. Given previous suggestions of enhanced sensitivity to negative feedback in patients with OCD (*Apergis-Schoute et al., 2024*, *Becker et al., 2014*; *Kanen et al., 2019*), we predicted that the OCD group, as compared to the control group, would demonstrate greater trial-to-trial changes in movement time and a more negative center of the $p(\Delta T|\Delta R-)$ distribution. Additionally, we examined whether OCD participants would exhibit more irregular changes to $\Delta R-$ and $\Delta R+$ values, as reflected in a larger spread of the $p(\Delta T|\Delta R+)$ and $p(\Delta T|\Delta R-)$ distributions, compared to the control group.

The conditional probability distributions were separately fitted to subsamples of the data across *continuous* reward practices, splitting the total number of correct sequences into four bins. This analysis allowed us to assess changes in reward sensitivity and behavioral changes across bins of sequences (bins 1–4 by partitioning the total number of sequences, from the whole training, into four). We focused the analysis on the continuous reward schedule for two reasons: (1) changes in scores on this schedule are more obvious to the participants and (2) a larger number of trials in each subsample were available to fit the Gaussian distributions, due to performance-related reward feedback being provided on all trials.

We observed that participants speeded up their sequence duration more (negative changes in trial-wise $MT$) following a drop in scores, as expected (*Figure 5a*). Conducting a three-way analysis of variance (ANOVA) with reward change (increase, decrease) and bin (1:4, each bin denoting ~110 sequences) as within-subject factors, and group as between-subject factor, we found a significant main effect of reward ($F(504,1) = 319.383$, $p = 2.0 \times 10^{-16}$). This outcome indicated that, in both groups, participants reduced $MT$ differently as a function of the change in reward. There was also a significant main effect of bin ($F(504,3) = 19.583$, $p = 5.06 \times 10^{-12}$), such that participants sped up their sequence performance over practices. The main effects are illustrated in *Figure 5b*. There was no significant main group effect ($F(504,1) = 1.099$, $p = 0.2951$) and omitting the group factor from the full model using a BF ANOVA improved the model moderately ($BF = 6.08$, moderate evidence in support of the model with the main group effect removed relative to the full model). Thus, both OCD and HV individuals introduced comparable changes in $MT$ overall during training.

In addition, there was a significant interaction between reward and bin in predicting the trial-to-trial changes in movement time ($F(504,3) = 3.652$, $p = 0.0126$). This outcome suggested that the relative improvement in $MT$ over sequences depended on whether the reward increased or decreased from the previous trial. To explore this interaction effect further, we conducted a dependent-sample pairwise $t$-test on $MT$, after collapsing the data across groups. In each sequence bin, participants speeded up $MT$ more following a drop in scores than following an increment, as expected (corrected $p_{FDR} = 2 \times 10^{-16}$).

On the other hand, assessing the effect of bins separately for each level of reward, we observed that the large sensitivity of normalized $MT$ changes to reward decrements was attenuated from the first to the second bin of practice (corrected $p_{FDR} = 0.00034$, significant attenuation for pairs 1–2; dependent-sample t-tests between consecutive pairs of bins). No further changes over practice bins were observed ($p > 0.53$, no change for pairs 2–3, 3–4). Similarly, the sensitivity of $MT$ changes to

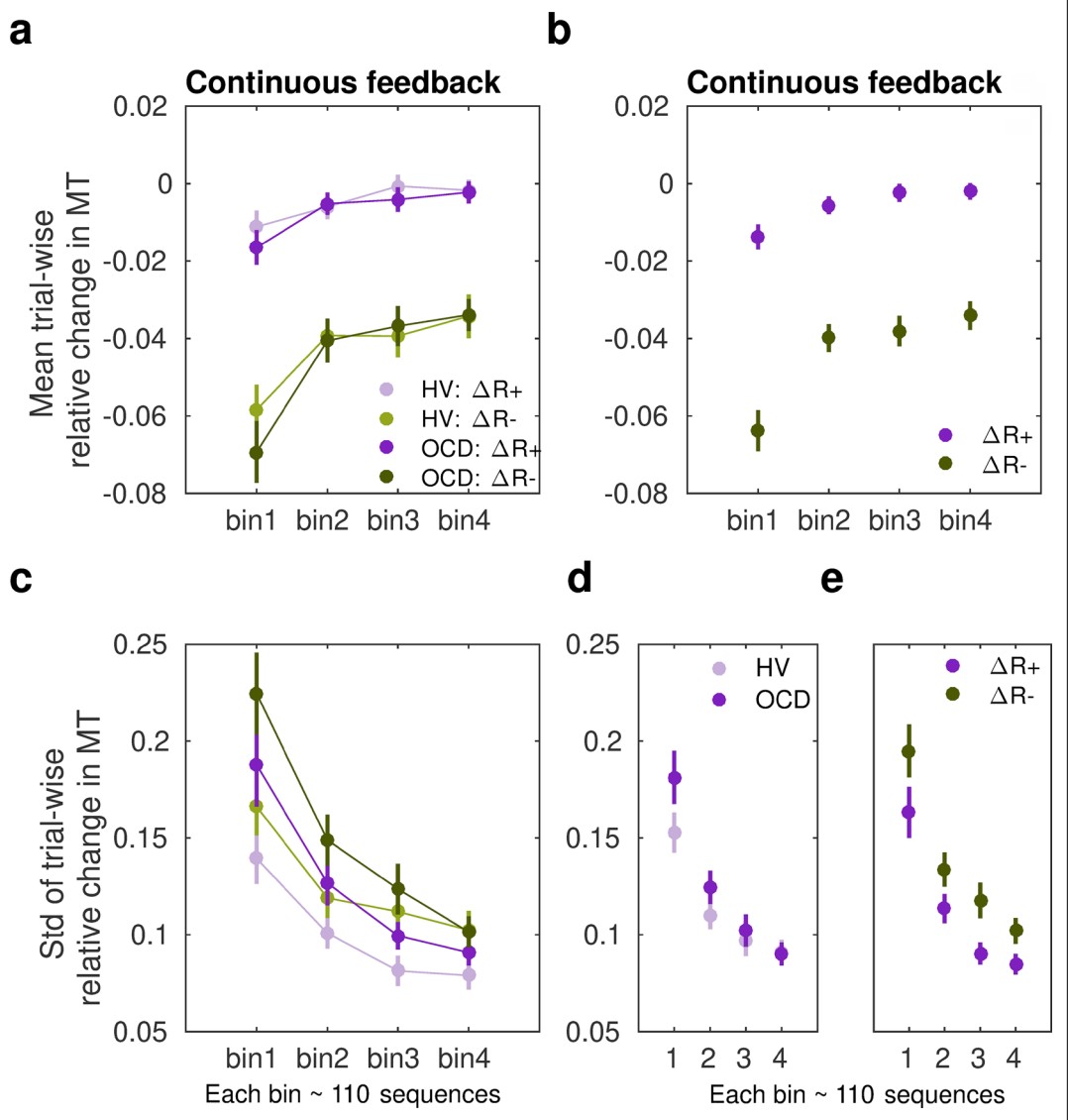

**Figure 5.** Sensitivity of movement time to changes in reward in the continuous reward schedule. (**a**) Mean normalized change in movement time (*MT*, ms) from trial *n* to n+1 following an increment (*ΔR+*, in purple) or decrement (*ΔR–*, in green) in scores at *n*. The change in movement time trial to trial was normalized with the baseline value on the initial trial *n*: $\Delta MT^{(n+1)} = (MT^{(n+1)} - MT^{(n)})/MT^{(n)}$. This relative change index is therefore adimensional. The dots represent mean *MT* changes (error bars denote SEM) in each bin of correctly performed sequences, after partitioning all correct sequences into four subsets, and separately for obsessive-compulsive disorder (OCD, N = 32) (dark colors) and healthy volunteers (HV, N = 33) (light colors). (**b**) Both groups of participants speeded up their sequence performance more following a drop in scores (main effect of reward, $p = 2 \times 10^{-16}$; 2×4: reward × bin analyses of variance [ANOVA]); yet this acceleration was reduced over the course of practiced sequences (main bin effect, $p = 5.06 \times 10^{-12}$). (**c**) Same as (a) but for the spread (std) of the *MT* change distribution (adimensional). (**d–e**) Illustration of the main effect of group (**d**) $p = 9.93 \times 10^{-6}$ and reward (**e**) $p = 4.13 \times 10^{-5}$ on std. Each bin depicted in the plots (x-axis) contains around 110 correct sequences on average (further details in Appendix 1: *Sample size for the reward sensitivity analysis*).

reward increments – consistently smaller – did only change from bin 1 to bin 2 ($p_{FDR}$ = 0.01092; no significant changes for pairs 2–3 and 3–4, $p > 0.31670$).

Overall, these findings indicate that both OCD and HV participants exhibited an acceleration in sequence performance following a decrease in scores (main effect). Furthermore, the sensitivity to score decrements or increments was reduced as participants approached automaticity through

repeated practice. Crucially, however, the increased sensitivity to reward decrements relative to increments persisted throughout the practice sessions in both groups.

Assessment of the std ($\sigma$) of the Gaussian distributions $p(\Delta T|\Delta R-)$ and $p(\Delta T|\Delta R+)$ in the continuous reward condition (*Figure 5c*) with a similar three-way ANOVA revealed a significant main effect of group ($F (504,1) = 19.928$, $p = 9.93 \times 10^{-6}$). As shown in *Figure 5d*, the std ($\sigma$) of the distribution of trial-to-trial *MT* changes was smaller in HV than in OCD. In addition, we observed a significant change over bins of sequences in $\sigma$, and independently of the group or reward factors (main effect of bin, $F (504,3) = 39.078$, $p = 2 \times 10^{-16}$). This outcome reflected that over the course of training, both groups exhibited less variable changes in MT in response to both reward increments and decrements, in line with improvements in skill learning (*Wolpert et al., 2011*). Reward also modulated $\sigma$, with $\Delta R-$ being associated with a more variable distribution of behavioral changes than $\Delta R+$(main effect of reward, $F (504,1) = 17.110$, $p = 4.13 \times 10^{-05}$). No interaction effects were found (there was moderate to strong evidence that removing any of the possible interaction effects improved the model: *BF* ranged from 5.67 to 41.3). Control analyses demonstrated that the group, reward, or bin effects were not confounded by differences in the size of the subsamples used for the Gaussian distribution fits (not shown; Appendix 1 -*Sample size for the reward sensitivity analysis*).

## Sensitivity of IKI consistency (*C*) to reward

To further explore the potential impact of reward changes on the previously reported group effects on automaticity, we quantified the trial-by-trial behavioral changes in IKI consistency (represented by *C*) in response to changes in reward scores relative to the previous trial. Note that a smaller *C* indicates a more reproducible IKI pattern trial to trial. As for $\Delta MT$, we normalized the index *C* (termed *normC* to avoid confusion with the main analysis on *C*) with the baseline IKI values on the previous trial $n$

$$normC = \sum_{k=1}^{5} \left| (t_{k,n+1} - t_{k,n})/t_{k,n} \right|, \tag{8}$$

where $k$ is the inter-keystroke response interval and $n$ is the sequence trial number. During continuous reward practices, both patients and healthy controls exhibited an increased consistency of IKI patterns trial to trial across bins of correct sequences (decreased *normC*, *Equation 8*, *Figure 6a*; main effect of bin on the center of the Gaussian distribution, $F (497, 3) = 4.188$, $p = 0.00607$; three-way ANOVA). Performance in OCD and HV, however, differed with regard to how reproducible their timing patterns were (main effect of group, $F (497, 1) = 8.130$, $p = 0.00454$). The timing patterns were less consistent trial to trial in OCD, relative to HV (*Figure 6b*, left panel). Moreover, the IKI consistency improved more (smaller *normC*) following reward increments than after decrements, as shown in *Figure 6b* (right panel; main reward effect, $F (497, 1) = 23.283$, $p = 1.86 \times 10^{-6}$). No significant interaction effects between factors were found (BF analysis demonstrated that when any of the interaction effects among factors was removed from the ANOVA design, there was moderate to strong evidence that the model improved: BF in range from 6.83 to 14.1). Accordingly, although OCD participants exhibited an attenuated IKI consistency in their performance relative to HV, the main effects of reward and bins of sequences were independent of the group.

Regarding the spread of the $p(\Delta T|\Delta R)$ distributions, we found a significant main effect of bin factor ($F (504,3) = 23.350$, $p = 3.63 \times 10^{-14}$; *Figure 6c and d*). These outcomes suggest that the $\sigma$ of the Gaussian distribution for *normC* values was reduced across bins of practiced sequences. There was only a trend for a significant main effect of group ($F (504,1) = 3.412$, $p = 0.0653$) and no main effect of reward ($F (504,1) = 2.327$, $p = 0.1278$). These non-significant effects were explored further using BF. This analysis provided anecdotal and moderate evidence that omitting either the group or reward effects was beneficial to the model (*BF* = 1.98 for removing group, *BF* = 3.20 for removing reward). We did not observe any interaction effect either (BF values increased moderately to strongly when any of the interaction effects among factors was removed from the ANOVA design: BF ranged from 4.89 to 43.3). The results highlight that over the course of training participants' normalized IKI consistency values stabilized, and this effect was not observed to be modulated by group or reward factors. Similarly to the *MT* analyses, the sensitivity analyses of *normC* were not influenced by differences in the size of the subsamples used for the $\Delta R+$ and $\Delta R-$ Gaussian distribution fits (Appendix 1 - *Sample size for the reward sensitivity analysis*).

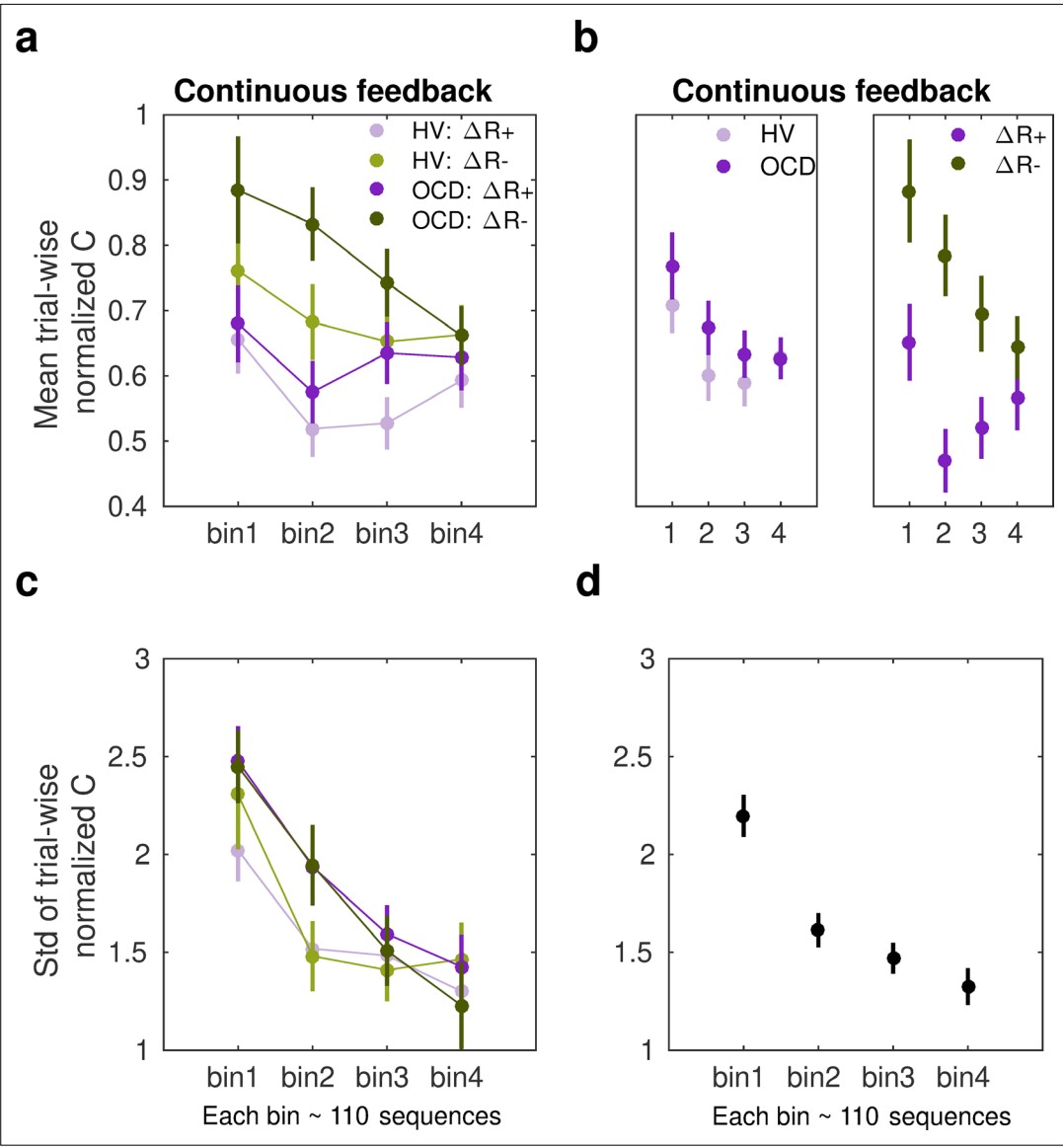

**Figure 6.** Sensitivity of normalized inter-keystroke interval (IKI) consistency (*normC*) to reward changes in the continuous schedule. (**a**) The mean normalized change in trial-to-trial IKI consistency (*normC*, *Equation 8* adimensional) across bins of correct sequences is shown, separately for each group (obsessive-compulsive disorder [OCD, N = 32]: dark colors; healthy volunteers [HV, N = 33]: light colors) and for reward increments (*ΔR+*, purple) and decrements (*ΔR–*, green). The dots represent the mean value, while the vertical bars denote SEM. (**b**) Illustration of the main effect of group (left panel; *p* = 0.00454) and type of reward change (right panel; *p* = 1.86 × 10$^{-6}$). (**c**) Same as (**a**) but for the std of the distribution of IKI consistency changes, *normC*, adimensional. (**d**) The panel displays the main effect of bin (*p* = 3.63 × 10$^{-14}$) on the std. Black denotes the average (SEM) across reward and group levels. Each bin depicted in the plots (*x*-axis) contains 110 correct sequences on average (see Appendix 1: *Sample size for the reward sensitivity analysis*).

## Phase B: Tests of action sequence preference and re-evaluation

Once the month-long app training was completed, participants attended a laboratory session to conduct additional behavioral tests aimed at assessing preference for familiar versus novel sequences (experiments 2 and 3) including a re-evaluation test to assess ability to adapt to environmental changes (experiment 3 only). Below we briefly describe these two experiments and report the results. See Materials and methods and Table 3 for a more detailed description of the tasks. Since these follow-up tests required observing additional stimuli *while* performing the action sequences, it was impractical to use participant's individual iPhones to simultaneously present the task stimuli and be an interface

to play the action sequences. We therefore used a 'Makey-Makey' device to connect the testing laptop (presenting the task stimuli) to four playdough keys arranged on a table (used as an interface for action sequence input). This device ensured precise key registration and timing. The playdough keys matched the size of those on the participants' iPhones used for the 1-month training. Participants practiced the action sequences in this new setup until they were comfortable. Hence, the transition to a non-mobile/laboratory context was conducted with great care. These tasks were conducted in a new context, which has been shown to promote re-engagement of the goal system (*Bouton, 2021*).

### Experiment 2: Preference for familiar versus novel action sequences

This experiment tests the hypothesis stated in the outline, that the trained action sequence gains intrinsic/rewarding properties or value. We used an *explicit preference task*, assessing participants' preferences for familiar (hypothetically habitual) sequences over goal-seeking sequences. We assume that if the trained sequences have acquired rewarding properties (e.g. anxiety relief, or the inherent gratification of skilled performance or routine), participants would express a greater preference to 'play' them, even when alternative easier sequences are offered (i.e. goal-seeking sequences).

After reporting which app sequence was their preferred, participants started the *explicit preference task*. On each trial, they were required to select and play one of two sequences. The two possible sequences were presented and identified using a corresponding image. Participants had to choose which one to play. There were three conditions, each comprising a specific sequence pair: (1) app preferred sequence *versus* app non-preferred sequence (*control condition*); (2) app preferred sequence *versus* any 6-move sequence (*experimental condition 1*); (3) app preferred sequence *versus* any 3-move sequence (*experimental condition 2*). The app preferred sequence was their preferred putative habitual sequence while the 'any 6'- or 'any 3'-move sequences were the goal-seeking sequences. These were considered less effortful for two reasons: (1) they could comprise any key press of participant's choice, even repeated presses of the same key (six or three times, respectively), and (2) they allowed for variations in key combinations each time the 'any-sequence' was input, rather than a fixed sequence on every trial. The conditions (15 trials each) were presented sequentially but counterbalanced among participants. See Materials and methods and *Figure 7a* for further details.

A Kruskal-Wallis *H* test indicated a main effect of condition ($H = 23.2$, $p < 0.001$) but no group ($p = 0.36$) or interaction effects ($p = 0.72$) (*Figure 7b*). Dunn's post hoc pairwise comparisons revealed that *experimental condition 2 (app sequence versus any three sequence)* was significantly different from *control condition* ($p_{FDR} < 0.001$) and from *experimental condition 1 (app sequence versus any six sequence)* ($p_{FDR} = 0.006$). No differences were found between the latter two conditions ($p = 0.086$). Bayesian analysis further provided moderate evidence in support of the absence of main effects of group ($BF = 0.129$) and interaction ($BF = 0.054$). These results denote that both groups evaluate the trained app sequences as being equally attractive as the alternative novel-but-easier sequence when of the same length (*Figure 7b*, middle plot). However, when given the option to play an easier-but-shorter sequence (in *experimental condition 2*), both groups significantly preferred it over the app familiar sequence (*Figure 7b*, right plot). A positive correlation between COHS and the app sequence choice (*Pearson r* = 0.36, *p* = 0.005) showed that those participants with greater habitual tendencies had a greater propensity to prefer the trained app sequence under this condition.

Given the high variance of participants' choices on this preference task, particularly in the experimental conditions, and the findings reported below related to the mobile-app performance effect on symptomatology, we further conducted an exploratory Dunn's post hoc test splitting the OCD group into two subgroups based on their Yale-Brown Obsessive-Compulsive Scale (YBOCS) score changes after the app training: 14 patients with improved symptomatology (reduction in YBOCS scores) and 18 patients who remained stable or felt worse (i.e. respectively, same or increase in YBOCS scores). Patients with lowered YBOCS scores after the app training had significantly greater preference for the app trained sequence in both experimental conditions as compared to patients with same or increased YBOCS scores after the app training: *experimental condition 1* ($p_{FDR} = 0.015$, *Figure 7c*, left) and *experimental condition 2* ($p_{FDR} = 0.011$, *Figure 7c*, right). In addition to this subgrouping analysis, we conducted a correlation analysis between changes in YBOCS scores and patient preferences for the app sequences. This helped us determine whether patients who experienced greater changes in YBOCS scores tended to prefer the learned sequences, and vice versa. We observed a positive correlation, meaning that the higher the symptom improvement after the month training,

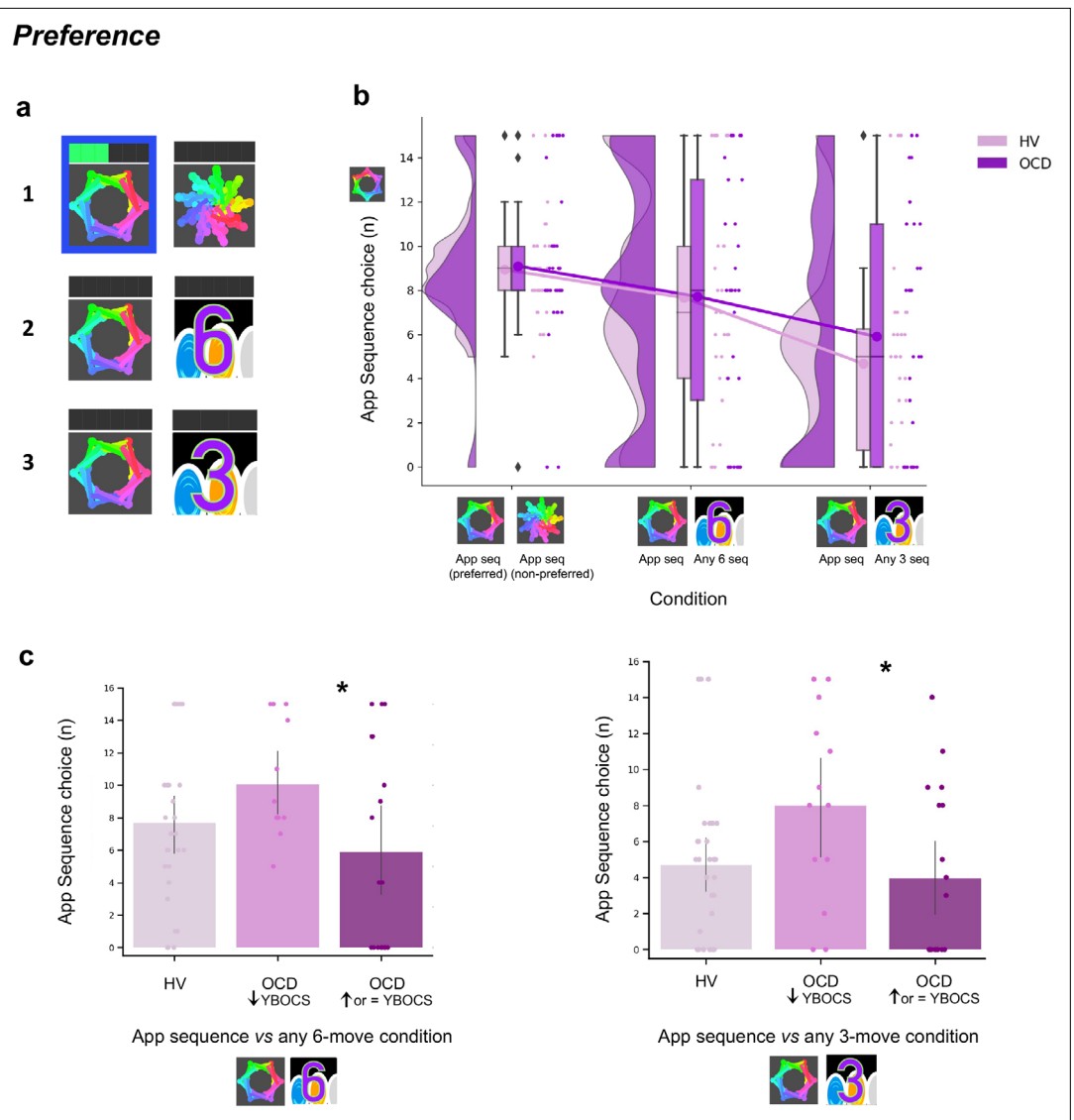

**Figure 7.** Preference for familiar versus novel action sequences. (**a**) Explicit preference task. Participants had to choose and play one of two given sequences. Once the choice was made, the image correspondent to the selected sequence was highlighted in blue. Participants then played the sequence. While playing it, the bar on top registered each move progressively lighting up in green. There were three conditions, each comprising a specific sequence pair: (1) app preferred sequence *versus* app non-preferred sequence (*control condition*); (2) app preferred sequence *versus* any 6-move sequence of participant's choice (*experimental condition 1*); (3) app preferred sequence *versus* any 3-move sequence of participant's choice (*experimental condition 2*). (**b**) No evidence for enhanced preference for the app sequence in either HV or OCD patients (HV: N = 33, OCD: N = 32). In fact, when an easier and shorter sequence is pitted against the app familiar sequence (right raincloud plot), both groups significantly preferred it (Kruskal-Wallis main effect of condition $H = 23.2$, $p < 0.001$). Left raincloud plot: control condition; middle raincloud plot: experimental condition 1; right raincloud plot: experimental condition 2. *Y*-axis depicts the number of app sequence choices (15 choice trials maximum). Connected lines depict mean values. (**c**) Exploratory analysis of the preference task following up unexpected findings on the mobile-app effect on symptomatology: re-analysis of the data conducting a Dunn's post hoc test splitting the OCD group into two subgroups based on their YBOCS change after the app training (14 patients with improved symptomatology [reduced YBOCS scores] and 18 patients who remained stable or felt worse [i.e. respectively unchanged or increased YBOCS scores]). Patients with reduced YBOCS scores after the app training had significantly higher preference to play the app sequence in both experimental conditions (left panel: $p_{FDR} = 0.015^*$; right panel: $p_{FDR} = 0.011^*$). The bar plots represent the sample mean and the vertical lines the confidence interval. Individual data points are included to show dispersion in the sample. Abbreviations: YBOCS = Yale-Brown Obsessive-Compulsive Scale, HV = healthy volunteers, OCD = patients with obsessive-compulsive disorder.

the greater the preference for the familiar/learned sequence. This is particularly the case for the experimental condition 2, when subjects are required to choose between the trained app sequence and any 3-move sequence ($r_s$ = 0.35, $p$ = 0.04). A trend was observed for the correlation between the YBOCS score change and the preference for the app sequences in experimental condition 1 ($r_s$ = 0.30, $p$ = 0.09). In conclusion, most participants preferred to play shorter and easier alternative sequences, thus not showing a bias toward the trained/familiar app sequences. Contradicting our hypothesis, OCD patients followed the same behavioral pattern. However, some participants still preferred the app sequence, specifically those with greater habitual tendencies, including patients who improved their symptoms during the month training and considered the app training beneficial (see also below exploratory analyses of 'Mobile-app performance effect on symptomatology'). Such preference presumably arose because some intrinsic value may have been attributed to the trained action sequence.

## Experiment 3: Re-evaluation of the learned action sequence

In experiment 3, we employed a *two-choice appetitive learning task*. We modified the conditions by manipulating extrinsic feedback to assess participants' capacity to adopt a different response choice, after re-evaluating their options. By providing more value to alternative action sequences (as opposed to the previously automatized ones), participants were thus encouraged to reassess their choices and respond appropriately. Of note, we did not use a conventional goal devaluation procedure here, as this could possibly have disrupted the behavioral control of the sequences and thus invalidated the test.

On each trial, participants were required to choose between two 'chests' based on their associated reward value. Each chest depicted an image identifying the sequence that needed to be completed to be opened. After choosing which chest they wanted, participants had to play the specific correct sequence to open it. Their task was to learn by trial and error which chest would give them more rewards (gems), which by the end of the experiment would be converted into real monetary reward. There was no penalty for incorrectly keyed sequences because behavior was assessed based on participants' choice regardless of the sequence accuracy.

Four chest-pairs (conditions, 40 trials each) were tested (see *Figure 8a* and Materials and methods for detailed description of each condition): three conditions pitted the trained/familiar app sequence against alternative sequences of higher monetary outcomes (given by variable amount of reward that did not overlap [deterministic]). The fourth condition kept the monetary value equivalent for the two options (maintaining a probabilistic rather than deterministic contingency) but offered a significantly easier/shorter alternative sequence. This set up a comparison between the intrinsic value of the familiar sequence and a motor-wise less effortful sequence. The conditions were presented sequentially but counterbalanced among participants.

Both groups were highly sensitive to the re-evaluation procedure based on monetary feedback, choosing more often the non-app sequence, irrespective of the novelty of that sequence (*Figure 8b*, no group effects; $p$ = 0.210 and $BF$ = 0.742, anecdotal evidence supporting no main effect of group). However, when re-evaluation required motor effort (condition 4), participants were less inclined to choose the 'any 3' alternative, which is the sequence demanding less motor effort (Kruskal-Wallis main effect of condition: $H$ = 151.1, $p$ < 0.001). Moreover, OCD patients significantly favored the trained app sequence over HV (post hoc group × condition 4 comparison: $p$ = 0.04). In conclusion, following the month of training, both groups exhibited the ability to update their behavior based on monetary re-evaluation. Yet, OCD patients more frequently selected the familiar sequence, even when a less effortful and shorter alternative was available.

## Mobile-app performance effect on symptomatology: exploratory analyses

In a debriefing questionnaire, participants were asked to give feedback about their app training experience and how it interfered with their routine: (1) how stressful/relaxing the training was (rated on a scale from –100% highly stressful to 100% very relaxing); (2) how much it impacted their life quality (*Q*) (rated on a scale from –100% maximum decrease to 100% maximum increase in life quality). *Appendix 1—table 1* and *Appendix 1—figure 4* depict participants' qualitative and quantitative feedback. Of the 33 HV, 30 reported the app was neutral and did not impact their lives, neither

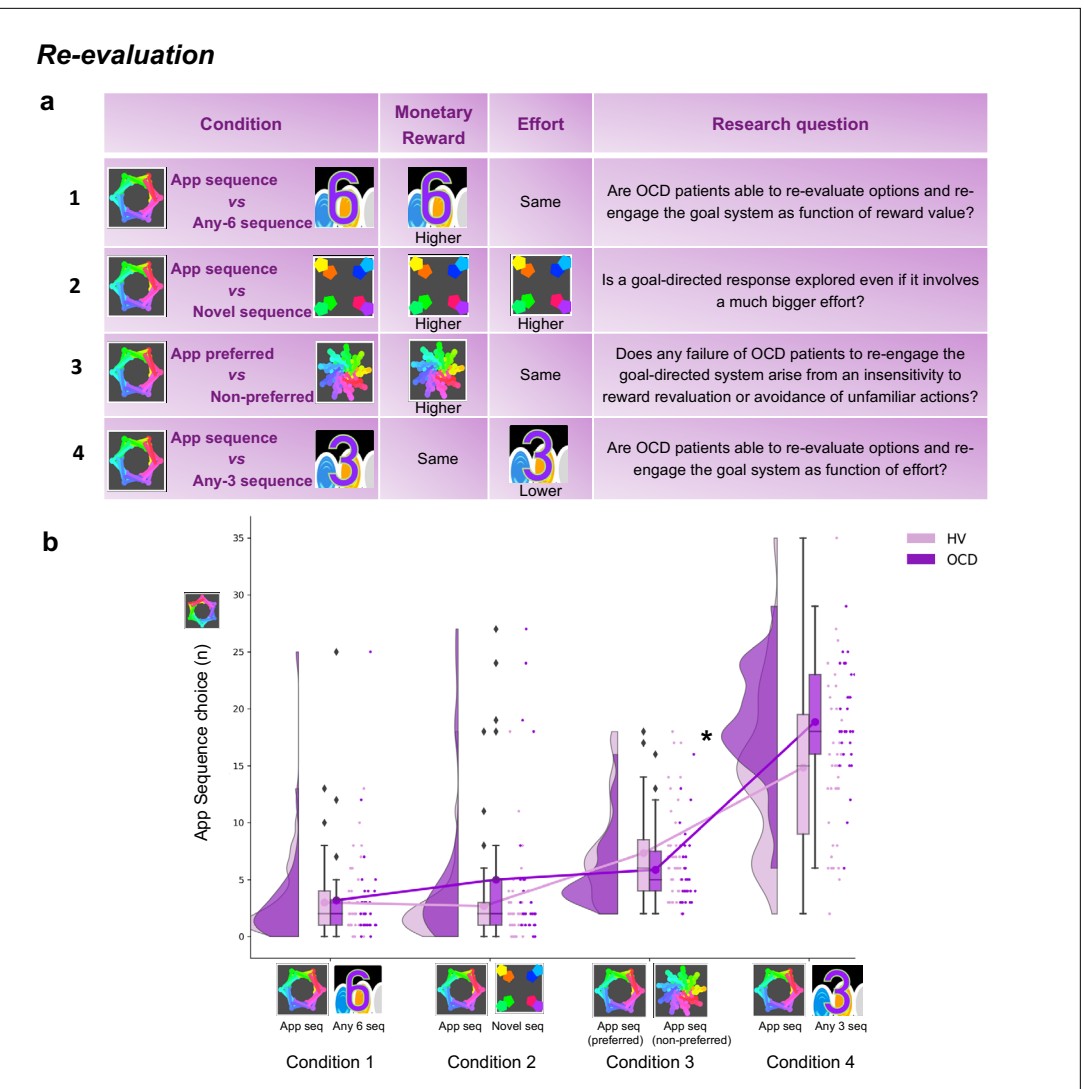

**Figure 8.** Re-evaluation procedure: two-choice appetitive learning task. (**a**) shows the task design. We tested four conditions, with chest-pairs corresponding to the following motor sequences: (1) app preferred sequence *versus* any 6-move sequence; (2) app preferred sequence *versus* novel (difficult) sequence; (3) app preferred sequence *versus* app non-preferred sequence; (4) app preferred sequence *versus* any 3-move sequence. The 'any 6-move' or 'any 3-move' sequences could comprise any key press of the participant's choice and could be played by different key press combinations on each trial. The 'novel sequence' (in 2) was a 6-move sequence of similar complexity and difficulty as the app sequences, but only learned on the test day (therefore, not overtrained). In conditions 1, 2, and 3, the preferred app sequence was pitted against alternative sequences of higher monetary value. In condition 4, the intrinsic value of the preferred app sequence was pitted against a motor-wise less effortful sequence (i.e. a shorter/easier sequence). Each condition addressed specific research questions, which are detailed in the right column of the table. (**b**) demonstrates the task performance per group and over the four conditions (HV: N = 33, OCD: N = 32). Both groups were able to adjust to the new contingencies and choose the sequences associated with higher monetary reward. When re-evaluation involved a motor effort manipulation, obsessive-compulsive disorder (OCD) patients chose the app sequence significantly more than healthy volunteers (HV) (* = $p < 0.05$) (condition 4). *Y*-axis depicts the number of app sequence chests chosen (40 trials maximum) and connected lines depict mean values.

positively nor negatively. The remaining 3 reported it as being a positive experience, with an improvement in their life quality (rating their life quality increase as 10%, 15%, and 60%). Of the 32 patients assessed, 14 unexpectedly showed improvement (*I*) in their OCD symptoms during the month as measured by the YBOCS difference, in percentage terms, pre-post training ($\bar{I} = 20 \pm 9\%$), 5 felt worse ($\bar{I} = -19 \pm 9\%$) and 13 remained stable during the month (all errors are standard deviations). Of the 14

who felt better, 10 directly related their OCD improvement to the app training (life quality increase: $\bar{Q}$ = 43 ± 24%). Nobody rated the app negatively. Of note, the symptom improvement was positively correlated with patients' habitual tendencies reported in the Creature of Habit questionnaire, particularly with the routine subscale (Pearson $r$ = 0.45, $p$ = 0.01) (*Figure 9a*, left). A three-way ANOVA test showed that patients who reported less obsessions and compulsions after the month training were the ones with more pronounced habit routines (group effect: $F$ = 13.7, $p$ < 0.001, *Figure 9a*, right). A strong positive correlation was also found between the OCD improvement reported subjectively as direct consequence of the app training and the OCI scores and reported habit tendencies (Pearson $r$ = 0.8, $p$ = 0.008; Pearson $r$ = 0.77, $p$ < 0.01, respectively) (*Figure 9b*): i.e., patients who considered the app somewhat beneficial were the ones with higher compulsivity scores and higher habitual tendencies. In HV, participants who also had greater tendency for automatic behaviors, regarded the app as more relaxing (Pearson $r$ = 0.44, $p$ < 0.01). However, such correlation between the self-reported relaxation measure attributed to the app and the COHS automaticity subscale was not observed in OCD

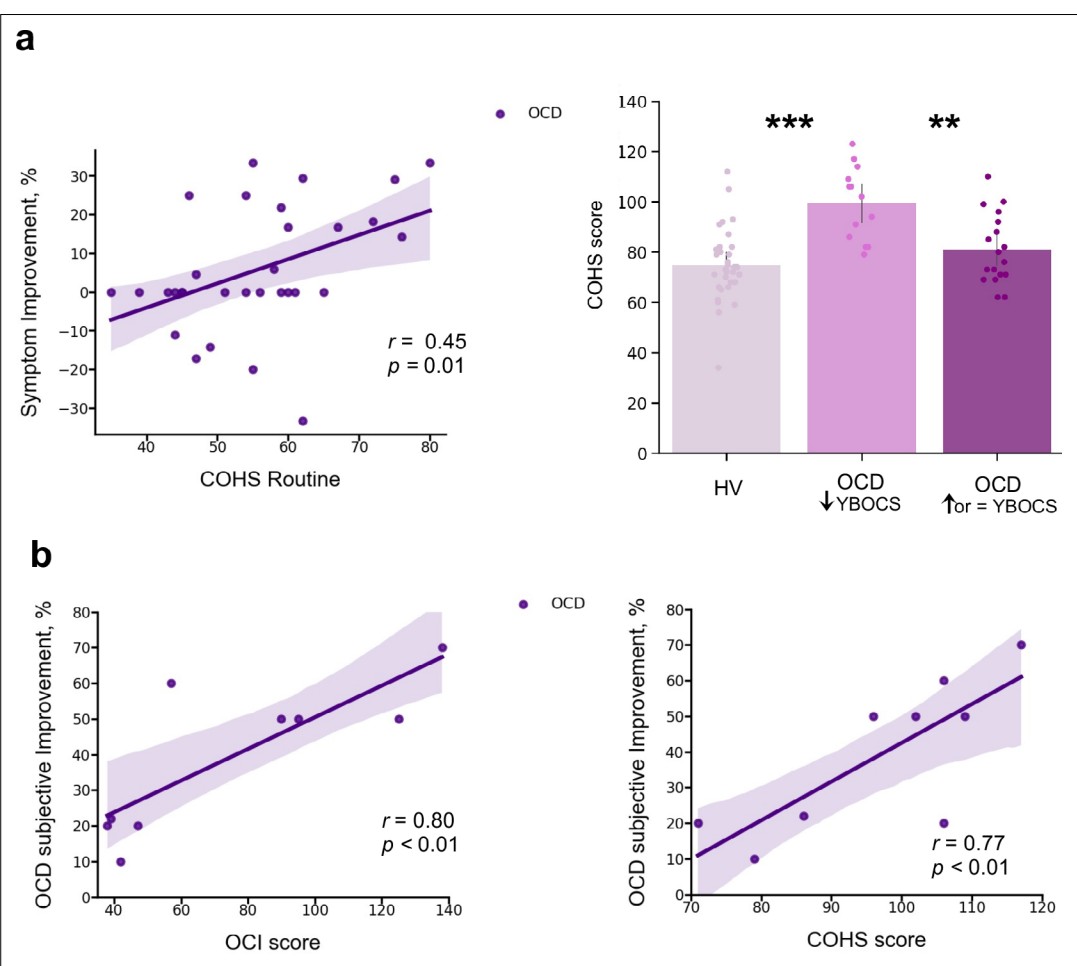

**Figure 9.** Mobile-app effect on symptomatology. (**a**) Left: Positive correlation between patients' routine tendencies reported in the COHS questionnaire and the symptom improvement (Pearson $r$ = 0.45, $p$ = 0.01), OCD sample size: N = 32. Symptom improvement was measured by the difference in YBOCS before and after app training. Right: Patients with greater improvement in their symptoms after the 1-month app training (N = 14) had greater habitual tendencies as compared to HV (N = 33) ($p$ < 0.001) and to patients who did not improve post-app training (N = 18) ($p$ = 0.002). The bar plot represents the sample means and the vertical lines the confidence interval. Individual data points are included to show dispersion in the sample. (**b**) OCD patients who related their symptom improvement directly to the app training (N = 9) were the ones with higher compulsivity scores on the OCI (Pearson $r$ = 0.8, $p$ = 0.008) (left) and higher habitual tendencies on the COHS (Pearson $r$ = 0.77, $p$ < 0.01) (right). Note that (**b**) has one missing patient because he did not complete the OCI scale and COHS. Abbreviations: OCI = Obsessive-Compulsive Inventory, COHS = Creature of Habit Scale, YBOCS = Yale-Brown Obsessive-Compulsive Scale, HV = healthy volunteers, OCD = patients with obsessive-compulsive disorder.

(*p* = 0.1). Finally, patients' symptom improvement did not correlate with how relaxing they considered the app training (*p* = 0.1) nor with the number of total practices performed during the month training period (*p* = 0.2).

We also checked whether the preferred app sequence, chosen by participants at the beginning of Phase B, was consistently the one that had yielded more reward during the app training (i.e. the continuously rewarded sequence). We found no evidence for this case: 54.5% of HV and 29% of the OCD sample considered the continuous sequence to be their preferred one, a non-statistically significant difference. This result suggests that participants' preference may not solely be linked to programmed reward. Other factors, such as the aesthetic appeal of, or ease of performing specific combinations of finger movements, may also influence overall preference.

## Other self-reported symptoms

In addition to the Creature of Habit findings, of the remaining self-reported questionnaires assessed (see Materials and methods), OCD patients also reported enhanced intolerance of uncertainty, elevated motivation to avoid aversive outcomes and higher perfectionism, worries and perceived stress, as compared to healthy controls (see *Table 1* for statistical results and *Figure 10* for overall summary).

**Table 1.** Self-reported measures on various scales measuring impulsiveness, compulsiveness, habitual tendencies, self-control, behavioral inhibition and activation, intolerance of uncertainty, perfectionism, stress, and the trait of worry.

|  | HV | OCD | Statistics | | |
|---|---|---|---|---|---|
|  | (*n* = 33) | (*n* = 32) | *t* | df | p |
| CPAS | 5.9 (4.0) | 14.2 (5.0) | –7.37 | 62 | <0.001[†] |
| COHS routine | 48.4 (9.4) | 55.7 (11.1) | –2.79 | 62 | 0.01* |
| COHS automaticity | 26.3 (8.2) | 32.9 (8.5) | –3.15 | 62 | <0.001[†] |
| COHS total | 74.8 (14.4) | 88.7 (16.7) | –3.56 | 62 | <0.001[†] |
| HSCQ | 50.7 (7.3) | 42.5 (8.5) | 4.17 | 62 | <0.001[†] |
| BIS | 17.5 (3.5) | 24.4 (2.7) | –8.81 | 61 | <0.001[†] |
| BAS reward responsibility | 15.9 (2.2) | 15.1 (2.5) | 1.25 | 61 | 0.22 |
| BAS drive | 10.0 (2.4) | 9.6 (2.6) | 0.66 | 61 | 0.51 |
| BAS fun seeking | 11.1 (1.9) | 9.7 (2.4) | 2.60 | 61 | 0.01* |
| Barratt total | 58.8 (8.4) | 65.0 (10.1) | –2.68 | 61 | 0.01* |
| Barratt attentional | 14.6 (4.1) | 19.8 (4.7) | –4.74 | 61 | <0.001[†] |
| Barratt motor | 21.2 (2.6) | 21.4 (3.2) | –0.23 | 61 | 0.82 |
| Barratt non-planning | 23.7 (3.3) | 24.6 (4.5) | –0.96 | 61 | 0.34 |
| IUS | 41.9 (10.0) | 87.3 (20.2) | –11.23 | 61 | <0.001[†] |
| SCS | 118.5 (21.4) | 118.3 (17.2) | 0.04 | 62 | 0.97 |
| FMPS | 70.3 (21.0) | 95.4 (21.4) | –4.73 | 62 | <0.001[†] |
| PSS | 13.7 (4.7) | 22.9 (5.1) | –7.51 | 62 | <0.001[†] |
| PSWQ | 37.9 (11.7) | 64.0 (11.0) | –9.20 | 62 | <0.001[†] |

HV, healthy volunteers; OCD, patients with obsessive-compulsive disorder; CPAS, Compulsive Personality Assessment Scale; COHS, Creature of Habit Scale; HSCQ, Habitual Self-Control Questionnaire; BIS, Behavioral Inhibition System; BAS, Behavioral Activation System; Barratt, Barratt Impulsiveness Scale; IUS, Intolerance of Uncertainty Scale; SCS, Self-Control Scale; FMPS, Frost Multidimensional Perfectionism Scale; PSS, Perceived Stress Scale; PSWQ, Penn State Worry Questionnaire. Standard deviations are in parentheses: mean (std). One patient and one healthy control missed a few questionnaires.

*= *p* < 0.05 level.

[†]= *p* < 0.001 level (two-tailed).

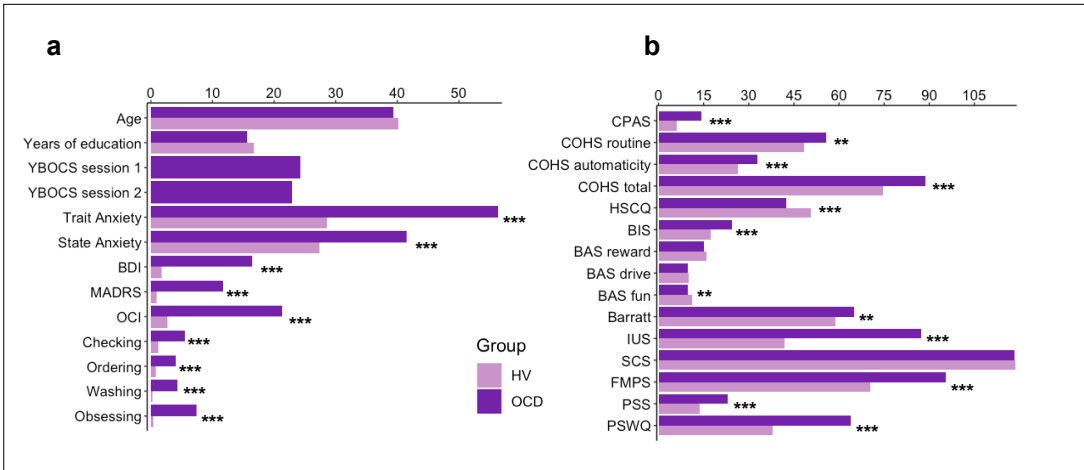

**Figure 10.** Participants' demographics, clinical characteristics and results from the self-reported questionnaires. (**a**) Participants' demographics and clinical characteristics (HV: N = 33, OCD: N = 32). (**b**) Between-group results from the self-reported questionnaires. Abbreviations: HV, healthy Volunteers; OCD, patients with obsessive-compulsive disorder; YBOCS, Yale-Brown Obsessive-Compulsive Scale; MADRS, Montgomery-Asberg Depression Rating Scale; STAI, The State-Trait Anxiety Inventory; BDI, Beck Depression Inventory; OCI, Obsessive-Compulsive Inventory; CPAS, Compulsive Personality Assessment Scale; COHS, Creature of Habit Scale; HSCQ, Habitual Self-Control Questionnaire; BIS, Behavioral Inhibition System; BAS, Behavioral Activation System; Barratt, Barratt Impulsiveness Scale; IUS, Intolerance of Uncertainty Scale; SCS, Self-Control Scale; FMPS, Frost Multidimensional Perfectionism Scale; PSS, Perceived Stress Scale; PSWQ, Penn State Worry Questionnaire. ** = $p < 0.01$, *** = $p < 0.001$.

## Discussion

This study investigated the roles of habits, their automaticity, and potential adjustments to environmental changes underlying compulsive OCD symptoms. We specifically focused on the habitual component of the associative dual-process model of behavior as applied to OCD and described in the Introduction. Using a self-report questionnaire (*Ersche et al., 2017*), we observed heightened subjective habitual tendencies in OCD patients across both the 'routine' and 'automaticity' domains, in comparison to controls.

Leveraging a novel smartphone tool, we real-time monitored the acquisition of two putative 'procedural' habits (six-element action sequences) in OCD patients and healthy participants over 30 days in their daily environments. Our analyses revealed heightened engagement with the app training among OCD patients; they enjoyed and practiced the sequences more than healthy participants without any explicit directive to do so. Initially, these patients performed the sequences more slowly and irregularly, yet they eventually achieved the same asymptotic level of automaticity and exhibited comparable 'chunking' (*Smith and Graybiel, 2016*) to controls. There were no discernible procedural learning deficits in patients, although their progression to automaticity was significantly slower than in healthy participants.

In a subsequent testing phase in a novel context, both groups adeptly transferred both trained action sequences to corresponding discriminative stimuli (visual icons). Furthermore, both cohorts were sensitive to re-evaluation when it pertained to monetary reward, demonstrating their ability to adapt behavior when facing environmental changes. However, when re-evaluation involved physical effort, OCD patients did not demonstrate the same adaptability and instead displayed a distinct inclination toward the already trained/familiar action sequence, presumably due to its inherent value. This effect was more pronounced in patients with higher habitual inclinations and compulsivity scores. Exploratory analysis revealed that patients with pronounced habitual inclinations and compulsivity scores were more likely to choose the familiar sequence. Moreover, when faced with a choice between the familiar and a new, less effort-demanding sequence, the OCD group leaned toward the former, likely due to its inherent value. These insights align with the theory of goal direction/habit imbalance in OCD (*Gillan et al., 2016*), underscoring the dominance of habits in particular settings where they might hold intrinsic value. This inherent value could hypothetically be associated with symptom

alleviation. Corroborating this, post-training feedback and a measured difference in the YBOCS scale pre- and post-training suggest many patients found the app therapeutically beneficial.

## Implications for the dual associative theory of habitual and goal-directed control

Rapid execution, invariant response topography, action chunking, and low cognitive load have all been considered essential criteria for the definition of habits (*Balleine and Dezfouli, 2019*; *Haith and Krakauer, 2018*). We have successfully achieved all these elements with our app using the criteria of extensive training and context stability, both previously shown to be essential to enhance formation and strengthening of habits (*Haith and Krakauer, 2018*; *Verplanken and Wood, 2006*). *Context stability* was provided by the tactile, visual, and auditory stimulation associated with the phone itself, which establishes a strong and similar context for all participants, regardless of their concurrent circumstances. *Overtraining* has been one of the most important criteria for habit development, and used by many as an operational definition on how to form a habit (*Dickinson et al., 1995*; *Haith and Krakauer, 2018*; *Tricomi et al., 2009*) (for a review, see *Balleine and O'Doherty, 2010*), despite current controversies raised by *de Wit et al., 2018*, on its use as an objective test of habits. A recent study has demonstrated though that even short overtraining (1 day) is effective at producing habitual behavior in participants high in affective stress (*Pool et al., 2022*), confirming previous suggestions for the key role of anxiety and stress on the behavioral expression of habits (*Dias-Ferreira et al., 2009*; *Hartogsveld et al., 2020*; *Schwabe and Wolf, 2009*). Here, we have trained a clinical population with moderately high baseline levels of stress and anxiety, with training sessions of a higher order of magnitude than in previous studies (*de Wit et al., 2018*; *Gera et al., 2022*). By all accounts our overtraining is valid: to our knowledge the longest overtraining in human studies achieved so far. All participants attained automaticity, exhibiting similar and stable asymptotic performance, both in terms of speed and the invariance in the kinematics of the motor movement.

We succeeded in achieving automaticity – which at a neural level is known to reliably engage the brain's habitual circuitry (*Ashby et al., 2010*; *Bassett et al., 2015*; *Graybiel and Grafton, 2015*; *Lehéricy et al., 2005*) – and fulfilled three of the four criteria for the definition of habits according to *Balleine and Dezfouli, 2019* (rapid execution, invariant topography, and chunked action sequences). However, we were not able to test the fourth criterion of resistance to devaluation. Therefore, we are unable to firmly conclude that the action sequences are habits rather than, for example, goal-directed skills. According to a very recent study, also employing an app to study habitual behavior, the criterion of devaluation resistance was shown to apply to a three-element sequence with less training (*Gera et al., 2022*). Thus, overtraining of our six-element sequence might also have achieved behavioral autonomy from the goal in addition to behavioral automaticity. While we did not employ the conventional goal devaluation test, it is possible that some experts may interpret our follow-up experiment 3 (the re-evaluation test) as a measure of *Balleine and Dezfouli, 2019*, fourth criterion, which defines habits as '*insensitive to changes in their relationship to their individual consequences and the value of those consequences*'. Consequently, they may conclude that the app-trained sequences exhibited some features of goal-directed behavior. While this interpretation holds merit, the logical conclusion is that the app-trained sequences encompass both habitual and goal-directed qualities. This aligns with contemporary perspectives on skilled/habitual sequences (*Du and Haith, 2023*).

Regardless of whether the trained action sequences are labeled as procedural habits or goal-directed motor skills, one must question why OCD patients preferred familiar sequences in specific situations, even when it seemed counterproductive (e.g. in the effort condition). This observation leads to the hypothesis that motivation for action sequences might include other factors besides explicit goals, such as monetary rewards. The apparent (intrinsic) therapeutic value of performing these sequences further blurs the attribution of a singular goal such as monetary reward to human action sequences. One implication of this analysis may be to consider that behavior in general is 'goal-directed' but may vary in the balance of control by external and internal goals. This perspective aligns with motor control theories that classify the successful completion of a motor action, in the spatio-temporal sense, as 'goal-related'. Hence, underlying any action sequence is possibly a hierarchy of objectives, ranging from overt rewards like money to intrinsic relief from an endogenous state (e.g. anxiety or boredom). In light of these insights, the dual associative process framework of behavioral control might be better understood in terms of the relative importance of extrinsic versus intrinsic

outcomes. Another possible formulation is that habits, which depend initially on cached or historically acquired rewarding action values, may not necessarily lose current value, but instead acquire alternative sources of value (*Hommel and Wiers, 2017*; *Kruglanski and Szumowska, 2020*; *O'Doherty, 2014*).

## Implications for understanding OCD symptoms

We observed a slower and more irregular performance in patients with OCD as compared to healthy participants at the beginning of training. This was expected given previous reports of visuospatial and fine-motor skill difficulties in patients with OCD (*Bloch et al., 2011*). However, despite this initial slowness, no procedural learning deficits were found in our patient sample. This finding is inconsistent with other implicit learning deficits previously reported in OCD using the serial reaction time (SRT) paradigm (*Deckersbach et al., 2002*; *Joel et al., 2005*; *Kathmann et al., 2005*; *Rauch et al., 2001*; *Rauch et al., 1997*). Nevertheless, this result aligns with recent studies demonstrating successful learning both in patients with OCD (*Soref et al., 2018*) and in healthy individuals with subclinical OCD symptoms (*Barzilay et al., 2022*) when instructions are given explicitly, and participants intentionally search for the underlying sequence structure. In fact, our task does not tap into memory processes as strongly as SRT tasks because we explicitly demonstrate the sequence to participants before they begin their 30-day training, which likely decreases demands on procedural learning.

Quantifying trial-to-trial behavioral changes in response to a decrease or increase in reward suggested that the slower progression toward automaticity observed in OCD patients might be related to their more inconsistent response to changes in feedback scores compared to healthy participants. The adjustments that OCD participants made to sequence duration after a score change were more variable (with a larger $\sigma$) than those made by healthy individuals. Additionally, the normalized consistency index was higher for the OCD group than for HV, indicating more fluctuating changes from trial to trial in IKI. Despite these group differences, we observed that in both samples the consistency of IKI patterns improved after reward increments. This observation contrasts with the more pronounced MT acceleration in both groups when faced with negative reward changes.

A heightened sensitivity to negative feedback within the motor domain has been documented in the general population, influencing initial motor improvements, while an increase in reward primarily boosts motor retention (*Abe et al., 2011*; *Galea et al., 2015*; *Pekny et al., 2015*; *van Mastrigt et al., 2020*). OCD individuals have also been shown to have an amplified sensitivity to negative feedback (*Becker et al., 2014*). Our findings indicate that decreased feedback scores affect sequence duration and IKI consistency in distinct ways. Specifically, reduced score feedback hampered automatization (reducing the IKI consistency, increasing *normC*), even though it generally had a positive effect on movement speed. This heightened responsiveness of *MT* (the rewarded variable) to decreased feedback scores is consistent with recent studies (*Abe et al., 2011*; *Galea et al., 2015*; *Pekny et al., 2015*; *van Mastrigt et al., 2020*). Our results, however, do not support differences in OCD and healthy individuals with regard to sensitivity to negative score changes, unlike previous work highlighting increased response switching after negative feedback, hyperactive monitoring systems, and amplified prediction errors in OCD (*Hauser et al., 2017*; *Marzuki et al., 2021*). One possible interpretation of these divergent results relates to the type of feedback. Previous work in OCD employed explicit negative feedback. In contrast, our participants received positive reward feedback, which increased or decreased trial by trial in the continuous reward schedule. The implicit nature of a reduced positive score, which is fundamentally different from overt negative feedback, might not elicit the same heightened sensitivity in OCD patients. They may primarily respond to explicit indications of failure or error, as opposed to subtle reductions in positive feedback. Another possibility is that the salient responses to negative feedback in OCD could be specific to the early stages of learning and may not persist after training for more than 1 hr or on subsequent days. Follow-up work will address these questions explicitly.

Considering the hypothetically greater tendency in OCD to form habitual/automatic actions described earlier (*Gillan et al., 2014*; *Voon et al., 2015*), we predicted that OCD patients would attain automaticity faster than healthy controls. This was not the case. In fact, the opposite was found. Since this was the first study to our knowledge assessing action sequence automatization in OCD, our contrary findings may confirm recent suggestions that previous studies were tapping into goal-directed behavior rather than habitual control per se (*Gillan et al., 2015b*; *Vaghi et al., 2019*; *Zwosta*

*et al., 2018*) and may therefore have inferred enhanced habit formation in OCD as a defaulting consequence of impaired goal-directed responding. On the other hand, we are describing here two potential sources of evidence in favor of enhanced habit formation in OCD. First, OCD patients show a bias toward the previously trained, apparently disadvantageous, action sequences. In terms of the discussion above, this could possibly be reinterpreted as a narrowing of goals in OCD (*Robbins et al., 2019*) underlying compulsive behavior, in favor of its intrinsic outcomes. Second, OCD patients self-reported greater habitual tendencies in both the 'routine' and 'automaticity' subscales. Previous studies have reported that subjective habitual tendencies are associated with compulsive traits (*Ersche et al., 2019*; *Wuensch et al., 2022*) and act, in addition to cognitive inflexibility, as a predictor of subclinical OCD symptomatology in healthy populations (*Ramakrishnan et al., 2022*). There is an apparent discrepancy between self-reported 'automaticity' and the objective measure of automaticity we provided. This may result from a possible mis-labeling of this factor in the Creature of Habit questionnaire, where many of the relevant items indicate automatic S-R elicitation by situational triggering stimuli rather than motor topographic features of the behavior (e.g. '*when walking past a plate of sweets or biscuits, I can't resist taking one*').

Finally, we also expected that OCD patients would show a greater resistance than controls in adjusting their behavior when the extrinsic relative value of the trained familiar sequences is diminished, in the re-evaluation procedure. Our findings show that this is partially the case, depending on the type of reward considered. Although we showed that all participants, including OCD patients, were apparently goal-directed in terms of gaining money this was not so clear when goal re-evaluation involved the physical effort expended. In this latter manipulation, participants were less goal-oriented and OCD patients preferred to perform the longer, familiar, to the shorter, novel sequence, thus exhibiting significantly greater habitual tendencies, as compared to controls. Such group differences may be driven hypothetically by the intrinsic value associated with the automatic sequences.

## Possible beneficial effect of action sequence training on OCD symptoms as habit reversal therapy

OCD patients engaged significantly more with the Motor Sequencing App and enjoyed it more than HV. Additionally, the patients more prone to routine habits (COHS), with higher OCI scores, and who additionally showed a preference for familiar sequences (possibly by attributing to them intrinsic value), found the use of the app beneficial, exhibiting symptomatic improvement based on the YBOCS. One hypothesis for the therapeutic potential of this motor sequencing training is that the trained action sequences may disrupt OCD compulsions, either via 'distraction' or habit 'replacement', by engaging the same neural 'habit circuitry'. This habit 'replacement' hypothesis is in line with successful interventions in Tourette syndrome (*Hwang et al., 2012*), tic disorders (*Bate et al., 2011*), and trichotillomania (*Morris et al., 2013*).

### Limitations

As mentioned above, we were unable to employ the often-mooted 'gold standard' criterion of resistance to devaluation because it would have invalidated the subsequent tests. This meant that we were unable to conclusively define the trained action sequences as habitual according to the full set of *Balleine and Dezfouli, 2019* criteria, although they satisfied other important criteria such as automatic execution, invariant response topography, action chunking and low cognitive load. Nevertheless, the utility of the devaluation criterion has been questioned especially when applied to human studies of habit learning. This is because achieving devaluation can be difficult given that human behavior has multiple goals, some of which may be implicit, and thus difficult to control experimentally, as well as being subject to great individual variation. In fact recent analyses of habitual behavior have not employed devaluation or revaluation as a criterion (*Du and Haith, 2023*). That study ascertains habits using different criteria and provides supporting evidence for trained action sequences being understood as skills, with both goal-directed and habitual components.

Although we found a significant preference for the trained action sequence in OCD patients in the condition where it was pitted against a simpler and shorter motor sequence, as compared to the monetary discounting condition, the reason for this difference is not immediately obvious. However, it may have arisen because of the nature of the contingencies inherent in these choice tests. Specifically, the 'monetary discounting' condition involved a simple deterministic choice between the two alternatives,

which should readily be resolved in favor of the option associated with the greater, non-overlapping, range of rewards provided (e.g. 1–7 versus 8–15 gems). In contrast, in the 'effort discounting' condition, the reward ranges for the two options were equivalent (e.g. 1–7 gems), which raised uncertainty concerning which of the chosen sequences was optimal. The probabilistic constraint over this choice may therefore account for the greater sensitivity of the task in highlighting preference in OCD, given the greater susceptibility of such patients to uncertainty (*Pushkarskaya et al., 2015*).

Finally, some of the conclusions relating to the effects of OCD severity on sequence preference without feedback were based only on a post hoc exploratory analysis. Specifically, only those patients with higher compulsivity (OCI) and COHS scores exhibited this preference, therefore consistent with the hypothesis described above of the importance of intrinsic value of the habitual sequence to the development of compulsions. Evidence of this intrinsic value was provided by the greater engagement with, and therapeutic findings for, the app training in these patients. However, the latter effect needs to be confirmed in a registered clinical trial in a controlled manner, which is ongoing.

## Conclusion

We employed a battery of behavioral tasks designed to investigate two key hypotheses of the goal/habit imbalance theory of compulsion, specifically pertaining to enhanced habit formation and automaticity and impaired goal re-evaluation in individuals with OCD. Our findings did not support greater habit formation nor heightened automaticity in patients with OCD. Moreover, evidence for patients' ability to adapt behavior when facing environmental changes was mixed. In certain contexts, OCD patients were able to behaviorally re-adjust (e.g. when reward is monetary) but in others (e.g. when involving motor effort) patients demonstrated a distinct augmented inclination to perform their trained/familiar action sequences, attributing higher intrinsic value to them. Interestingly, this preference was more pronounced in patients with higher compulsivity and habitual tendencies, who engaged significantly more with the motor habit-training app, reporting symptom relief after the experiment. This suggests a promising avenue for investigating the therapeutic potential of this application as a tool for habit reversal in the context of OCD.

## Materials and methods

### Participants

We recruited 33 OCD patients and 34 healthy individuals, matched for age, gender, IQ, and years of education. Two participants (one HV and one OCD) were excluded because they did not perform the minimum required training (i.e. two daily practices for a period of 30 days). Therefore, a total of 32 OCD patients (19 females) and 33 healthy participants (19 females) were included in the analysis. Most participants were right-handed (left-handed: four OCD and six HV). Participants' demographics and clinical characteristics are presented in *Table 2* and *Figure 10*.

Healthy individuals were recruited from the community, were all in good health, unmedicated, and had no history of neurological or psychiatric conditions. Patients with OCD were recruited through an approved advertisement on the OCD action website (https://ocdaction.org.uk/) and local support groups and via clinicians in East Anglia. All patients were screened by a qualified psychiatrist of our team, using the Mini International Neuropsychiatric Inventory (MINI) to confirm the OCD diagnosis and the absence of any comorbid psychiatric conditions. Patients with hoarding symptoms were excluded. Our patient sample comprised 6 unmedicated patients, 20 taking selective serotonin reuptake inhibitors (SSRIs), and 6 on a combined therapy (SSRIs+antipsychotic). OCD symptom severity and characteristics were measured using the YBOCS scale (*Goodman et al., 1989*), mood status was assessed using the Montgomery-Asberg Depression Rating Scale (MADRS) (*Montgomery and Asberg, 1979*) and Beck Depression Inventory (BDI) (*Beck et al., 1961*), anxiety levels were evaluated using the State-Trait Anxiety Inventory (STAI) (*Spielberger et al., 1983*), and verbal IQ was quantified using the National Adult Reading Test (NART) (*Nelson and Willison, 1982*). All patients included suffered from OCD and scored >16 on the YBOCS, indicating at least moderate severity. They were also free from any additional axis I disorders. General exclusion criteria for both groups were substance dependence, current depression indexed by scores exceeding 16 on the MADRS, serious neurological or medical illnesses, or head injury. All participants completed additional self-report questionnaires measuring:

a. Impulsiveness: Barratt Impulsiveness Scale (*Barratt, 1994*)

**Table 2.** Demographic and clinical characteristics of OCD patients and matched healthy controls.

| | HV | OCD | Statistics | | |
| --- | --- | --- | --- | --- | --- |
| | (n = 33) | (n = 32) | t | df | p |
| Gender ratio (male/female) | 14/19 | 13/19 | | | |
| Age | 40.2 (11.7) | 39.3 (12.5) | 0.29 | 63 | 0.77 |
| Years of education | 16.8 (3.4) | 15.6 (3.5) | 1.33 | 63 | 0.19 |
| Predicted verbal IQ | 117.8 (5.6) | 118.4 (4.6) | –0.43 | 63 | 0.67 |
| YBOCS session 1 | 0.0 | 24.3 (5.7) | – | – | – |
| YBOCS Obsessions session1 | 0.0 | 12.2 (3.0) | – | – | – |
| YBOCS Compulsions session1 | 0.0 | 11.8 (3.7) | – | – | – |
| YBOCS session 2 | 0.0 | 22.9 (6.6) | – | – | – |
| YBOCS Obsessions session2 | 0.0 | 11.6 (3.1) | – | – | – |
| YBOCS Compulsions session2 | 0.0 | 11.1 (4.2) | – | – | – |
| Trait Anxiety (STAI-T) | 28.6 (5.9) | 56.4 (8.6) | –15.11 | 63 | <0.001*** |
| State Anxiety (STAI-S) | 28.6 (5.9) | 56.4 (8.6) | –15.2 | 63 | <0.001*** |
| BDI | 1.7 (2.3) | 16.5 (9.4) | –8.72 | 62 | <0.001*** |
| MADRS | 0.9 (1.5) | 11.8 (6.2) | –9.88 | 63 | <0.001*** |
| OCI | 7.3 (9.1) | 68.4 (30.9) | –10.83 | 62 | <0.001*** |
| Checking | 0.9 (1.9) | 11.7 (9.4) | –6.5 | 62 | <0.001*** |
| Ordering | 0.7 (1.6) | 5.8 (3.3) | –7.92 | 62 | <0.001*** |
| Washing | 7.3 (9.2) | 66.0 (28.6) | –11.18 | 62 | <0.001*** |
| Doubting | 1.9 (2.7) | 13.6 (7.5) | –8.37 | 62 | <0.001*** |
| Obsessing | 1.1 (1.8) | 7.9 (4.0) | –8.82 | 62 | <0.001*** |

Abbreviations: OCD, patients with obsessive-compulsive disorder; HV, healthy volunteers; YBOCS, Yale-Brown Obsessive-Compulsive Scale; MADRS, Montgomery-Asberg Depression Rating Scale; STAI, The State-Trait Anxiety Inventory; BDI, Beck Depression Inventory; OCI, Obsessive-Compulsive Inventory. Standard deviations are in parentheses: mean (std). One patient missed the BDI and the OCI questionnaires. *** = $p < 0.001$ level (two-tailed).

b.  Compulsiveness: Obsessive Compulsive Inventory (*Foa et al., 1998*) and Compulsive Personality Assessment Scale (*Fineberg et al., 2007*)
c.  Habitual tendencies: Creature of Habit Scale (*Ersche et al., 2017*)
d.  Self-control: Habitual Self-Control Questionnaire (*Schroder et al., 2013*) and Self-Control Scale (*Tangney et al., 2004*)
e.  Behavioral inhibition and activation: BIS/BAS Scale (*Carver and White, 1994*)
f.  Intolerance of uncertainty (*Buhr and Dugas, 2002*)
g.  Perfectionism: Frost Multidimensional Perfectionism Scale (*Frost and Marten, 1990*)
h.  Stress: Perceived Stress Scale (*Cohen et al., 1983*)
i.  Trait of worry: Penn State Worry Questionnaire (*Meyer et al., 1990*).

All participants gave written informed consent prior to participation, in accordance with the Declaration of Helsinki, and were financially compensated for their participation. This study was approved by the East of England – Cambridge South Research Ethics Committee (16/EE/0465).

## Phase B: Tests of action sequence preference and re-evaluation
### Experiment 2: Explicit preference task
Participants observed, on each trial, two sequences identified by a corresponding image, and were asked to choose which one they wanted to play. Once the choice was made, the image correspondent

to the selected sequence was highlighted in blue. Participants then played the sequence. The task included 3 conditions (15 trials each). Each condition comprised a specific sequence pair: 2 experimental conditions pairing the app preferred sequence (putative procedural habit) with a goal-seeking sequence and 1 control condition pairing both app sequences trained at home. The conditions were as follows: (1) app preferred sequence *versus* app non-preferred sequence (*control condition*); (2) app preferred sequence *versus* any 6-move sequence (*experimental condition 1*); (3) app preferred sequence *versus* any 3-move sequence (*experimental condition 2*). The app preferred sequence was the putative habitual sequence and the 'any 6'- or 'any 3'-move sequences were the goal-seeking sequences because they are supposedly easier: they could comprise any key press of participant's choice (e.g. the same single key press repeatedly six or three times, respectively) and they could have same or different key press combinations every time the 'any-sequence' needed to be input. The conditions (15 trials each) were presented sequentially but counterbalanced among participants. See *Figure 7a* for illustration of the task.

## Experiment 3: Two-choice appetitive learning task

On each trial, participants were presented with two 'chests', each containing an image identifying the sequence that needed to be completed to be able to open the chest. Participants had to choose which chest to open and play the correct sequence to open it. Their task was to learn by trial and error which chest would give them more rewards 'gems', which by the end of the experiment would be converted into real monetary reward. If mistakes were made inputting the sequences, participants could simply repeat the moves until they were correct, without any penalty. Behavior was assessed based on participants' choice, regardless of the accuracy of the sequence. The task included 4 conditions (40 trials each), with chest-pairs correspondent to the following motor sequences (see also *Figure 8* for illustration of each condition):

- condition 1: app preferred sequence versus any 6-move sequence
- condition 2: app preferred sequence versus a novel (difficult) sequence
- condition 3: app preferred sequence versus app non-preferred sequence
- condition 4: app preferred sequence versus any 3-move sequence

As in the preference task described above, the 'any 6-move' or 'any 3-move' sequences could comprise any key press of participant's choice (e.g. the same single key press repeatedly six or three times, respectively) and could be played by different key press combinations on each trial. The novel sequence (in condition 2) was a 6-move sequence of similar complexity and difficulty as the app sequences, but only learned on the day, before starting this task (therefore, not overtrained). The training of this novel sequence comprised 40 trials only: sufficient to learn the sequence without overtraining. Initially lighted keys guided the learning (similarly to the app training). After the initial five trials, the lighted cues were removed, and participants were required to input the previously well-learned correct 6-move sequence. When an error occurred, the correct input key(s) lighted up on the following trial (a few milliseconds before participants made key presses), to remind participants of the correct sequence and help them consolidate learning of the novel sequence. In conditions 1, 2, and 3, higher monetary outcomes were given to the alternative sequences. To remove the uncertainty confound commonly linked to probabilistic tasks, conditions 1, 2, and 3 followed a deterministic nature: in all trials, the choice for the preferred app sequence was rewarded with smaller monetary outcomes (sampled from a random distribution between 1 and 7 gems) whereas the alternative

**Table 3.** Follow-up task instructions.

Explicit preference task

You will be given two sequences to choose from.
You can play either of them and switch as you go.
Select the sequences using the left and right pads and then play it

Two-choice appetitive instrumental task

In the following task, you will need to choose between two chests. Pick a chest using the left and right pads and play the matching sequence to open it. Open any chest you want. One of the chests may reward you more than the other. The more gems you get, the more money you will earn at the end of the task. Try to win as much as you can! You will receive your winnings at the end of the study.

option always provided higher monetary outcomes (sampled from a random distribution between 8 and 15 gems). Therefore, variable amount of reward that did not overlap was given (deterministic). Condition 4, on the other hand, kept the monetary value equivalent for the two options (maintaining a probabilistic rather than deterministic contingency) but offered a significantly easier/shorter alternative sequence. This set up a comparison between the intrinsic value of the familiar sequence and a motor-wise less effortful sequence. To prevent excessive memory load, which could introduce potential confounds, conditions were presented sequentially rather than intermixed, but the order was counterbalanced among participants (*Table 3*).

## Statistical analyses

Participant's characteristics and self-reported questionnaires were analyzed with $\chi^2$ and independent t-tests, respectively. The Motor Sequencing App automatically uploaded the data to a cloud-based database. This task enabled us to compare patients with OCD and HV in the following measures: training engagement (which included as primary output measures of the *total number of practices completed* and *app engagement* as defined as the number of sequences attempted, including both correct and incorrect sequences); procedural learning, automaticity development, sensitivity to reward (see definitions and description of data analyses in Results section); and training effects on symptomatology as measured by the YBOCS difference pre-post training. The Phase B experiments enabled further investigation of preference and re-evaluation strategies. The primary outcome was the number of choices.

Between-group analyses were conducted using Kruskal-Wallis *H* tests when the normality assumption was violated. Parametric factorial analyses were carried out with ANOVA. Our alpha level of significance was 0.05. On the descriptive statistics, main values are represented as median, and errors are reported as interquartile range unless otherwise stated, due to the non-Gaussian distribution of the datasets. When conducting several tests related to the same hypothesis, or when running several post-hoc tests following factorial effects, we controlled the FDR at level $p = 0.05$. Significant values after FDR control are denoted by $p_{FDR}$. Analyses were performed using Python version 3.7.6 and JASP version 0.14.1.0.

In the case of non-significant effects in the factorial analyses, we assessed the evidence in favor or against the full factorial model relative to the reduced model with BF (ratio *BF*full/*BF*restricted) using the bayesFactor toolbox (https://github.com/klabhub/bayesFactor, copy archived at *Klabhu, 2020*) in MATLAB. This toolbox implements tests that are based on multivariate generalizations of Cauchy priors on standardized effects (*Rouder et al., 2012*). As recommended by *Rouder et al., 2012*, we defined the restricted models as the full factorial model without one specific main or interaction effect. The ratio *BF*full/*BF*restricted represents the ratio between the probability of the data being observed under the full model and the probability of the same data under the restricted model. BF values were interpreted following *Andraszewicz et al., 2015*. The relationship between primary outcomes and clinical measures was calculated using a Pearson correlation.

The diurnal patterns of app use (*Figure 2b and c*) were assessed in each group using circular statistics (*Mardia, 1975*), with the 'circular' package in R (R version 4.3.1; 2023-06-16). This provided the group-level mean vector length and direction. To assess on the group level whether the daily practice data were uniformly distributed or, alternatively, oriented toward a specific time, we used a Rayleigh test (*Landler et al., 2021*; *Mardia, 1975*). We adapted code from *Galvez-Pol et al., 2022*. To test differences between two circular distributions (OCD, HV), we followed the recommendations of *Landler et al., 2021*, and employed the high-powered Watson's $U^2$ test, a non-parametric rank-based test (function watson.two.test in R).

## Code availability

The code for the main analyses is provided with this paper. It is available in the Open Science Framework, in the following link: https://osf.io/9xrdz/.

## Acknowledgements

This research was funded by the Wellcome Trust: a Sir Henry Postdoctoral Research Fellowship (Grant 204727/Z/16/Z) to PB and a Wellcome Trust Senior Investigator Award (Grant 104631/Z/14/Z) to TWR. For open access, the author has applied a CC BY public copyright licence to any Author Accepted

paper version arising from this submission. MB was supported by MHRUK and Angharad-Dodds Bursaries. AAM was supported as a research assistant funded by the aforementioned Wellcome Trust grant to TWR. We thank all participants for their contributions to this study. We would also like to acknowledge Dr. Sharon Morein-Zamir for fruitful brainstorm discussions on the tasks design.

## Additional information

### Competing interests

Naomi A Fineberg: NAF in the past three years has received research funding paid to her institution from the NIHR, COST Action and Orchard. She has received payment for lectures from the Global Mental Health Academy and for expert advisory work on psychopharmacology from the Medicines and Healthcare Products Regulatory Agency and an honorarium from Elsevier for editorial work. She has additionally received financial support to attend meetings from the British Association for Psycho-pharmacology, European College for Neuropsychopharmacology, Royal College of Psychiatrists, International College for Neuropsychopharmacology, World Psychiatric Association, International Forum for Mood and Anxiety Disorders, American College for Neuropsychopharmacology. In the past she has received funding from various pharmaceutical companies for research into the role of SSRIs and other forms of medication as treatments for OCD and for giving lectures and attending scientific meetings. Trevor W Robbins: TWR discloses consultancy with Cambridge Cognition and receives research grants from Shionogi & Co. He also has editorial honoraria from Springer Verlag and Elsevier. The other authors declare that no competing interests exist.

### Funding

| Funder | Grant reference number | Author |
|---|---|---|
| Wellcome Trust | 10.35802/204727 | Paula Banca |
| Wellcome Trust | 10.35802/104631 | Trevor W Robbins |
| Mental Health Research UK | | Marjan Biria |

The funders had no role in study design, data collection and interpretation, or the decision to submit the work for publication. For the purpose of Open Access, the authors have applied a CC BY public copyright license to any Author Accepted Manuscript version arising from this submission.

### Author contributions

Paula Banca, Conceptualization, Resources, Data curation, Software, Formal analysis, Supervision, Funding acquisition, Validation, Investigation, Visualization, Methodology, Writing – original draft, Project administration, Writing – review and editing, Data collection; Maria Herrojo Ruiz, Formal analysis, Writing – original draft; Miguel Fernando Gonzalez-Zalba, Formal analysis; Marjan Biria, Data collection; Aleya A Marzuki, Recruitment of participants; Thomas Piercy, Methodology; Akeem Sule, Patient recruitment; Naomi A Fineberg, Patient recruitment; Trevor W Robbins, Conceptualization, Supervision, Funding acquisition, Writing – review and editing

### Author ORCIDs

Paula Banca https://orcid.org/0000-0001-7617-3635
Maria Herrojo Ruiz http://orcid.org/0000-0001-8948-9444
Miguel Fernando Gonzalez-Zalba http://orcid.org/0000-0001-6964-3758
Trevor W Robbins https://orcid.org/0000-0003-0642-5977

### Ethics

All participants gave written informed consent prior to participation, in accordance with the Declaration of Helsinki, and were financially compensated for their participation. This study was approved by the East of England - Cambridge South Research Ethics Committee (16/EE/0465).

Reviewer #1 (Public review): https://doi.org/10.7554/eLife.87346.4.sa1

Author response https://doi.org/10.7554/eLife.87346.4.sa2

## Additional files

### Supplementary files
• MDAR checklist

### Data availability

The source data for all figures and analyses are provided with this paper. They are available in the Open Science Framework, in the following link: https://osf.io/9xrdz/.

The following dataset was generated:

| Author(s) | Year | Dataset title | Dataset URL | Database and Identifier |
|---|---|---|---|---|
| Banca P, Ruiz MH | 2023 | Action-sequence learning, habits and automaticity in OCD | https://osf.io/9xrdz/ | Open Science Framework, 9xrdz |

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

## Appendix 1

### Motor Sequencing App: additional task components and analysis

The Motor Sequencing App comprised additional components and analysis, which we describe and report below respectively. Once the minimum daily practice sessions were completed, we additionally asked participants to further conduct a short retention speed test, to assess that day's performance, and a switching practice session. In the five-trial retention speed session, participants were instructed to repeatedly tap a sequence as rapidly as possible while making as few errors as possible. The 10-trial switch session required switching between the two sequences in a pseudo-random order. The sequence to be played was cued by the respective associated picture. Speed and switch tests never received reward feedback (only the practice sessions). Finally, participants were asked to rate daily, on a percentage scale, the following two questions: (1) *How much did you enjoy playing this sequence?* and (2) *How confident are you that you know this sequence by heart?* This sequence of events (practice, speed, switch sessions, and ratings) happened every day.

### Confidence and enjoyment ratings

We conducted a mixed-design repeated measures ANOVA to investigate potential group differences (OCD versus HV) on ratings of confidence (*C*) and enjoyment (*E*) over time (4 weeks of app practice). We observed a main effect of time on confidence (*F* (3, 177) = 92.45, *p* < 0.001, $\eta_p^2$ = 0.19) but no Group (*p* = 0.61) or interaction (*p* = 0.95) effects (*Appendix 1—figure 1a*). This means that both groups significantly increased their confidence on their sequence knowledge over the course of the training. A Greenhouse-Geisser (*ε*) correction was applied given that sphericity was violated. Descriptive statistics are as follows (values provided as mean and standard deviation): HV: $C_{week1}$ = 64.03 ± 16.09%, $C_{week2}$ = 74.90 ± 17.52%, $C_{week3}$ = 80.14 ± 14.58%, $C_{week4}$ = 85.60 ± 12.90% and OCD: $C_{week1}$ = 61.66 ± 20.62%, $C_{week2}$ = 72.89 ± 18.40%, $C_{week3}$ = 78.55 ± 15.48%, $C_{week4}$ = 83.68 ± 14.23%. Regarding the enjoyment ratings, we found no significant main effects of group (*p* = 0.16), reward (*p* = 0.45), nor interaction effects (*p* = 0.25) (*Appendix 1—figure 1b*). Descriptive statistics are as follows (values provided as mean and standard deviation): HV: $E_{week1}$ = 56.98 ± 18.06%, $E_{week2}$ = 53.15 ± 24.72%, $E_{week3}$ = 51.61 ± 26.24%, $E_{week4}$ = 53.70 ± 30.28% and OCD : $E_{week1}$ = 59.23 ± 15.77, $E_{week2}$ = 61.19 ± 17.78%, $E_{week3}$ = 60.49 ± 22.93%, $E_{week4}$ = 64.41 ± 26.37%.

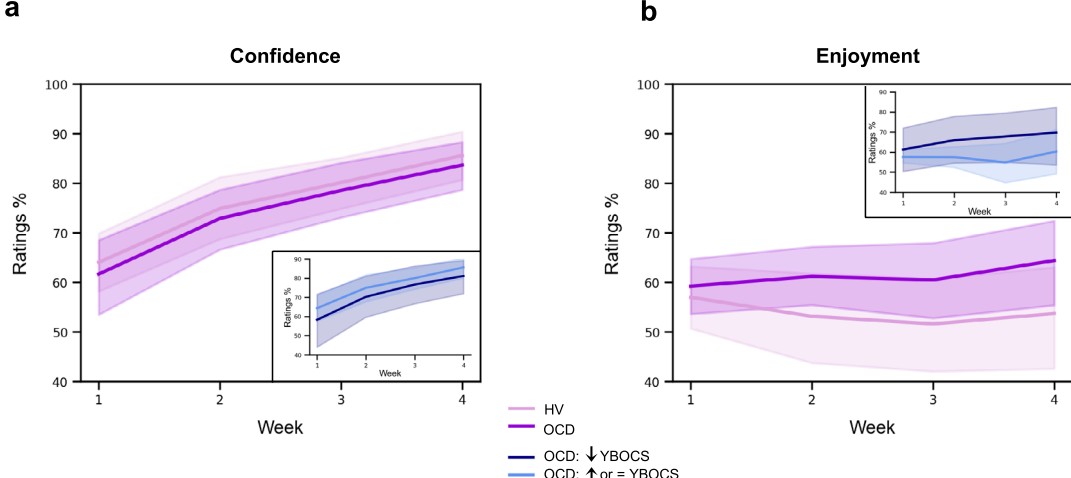

**Appendix 1—figure 1.** Confidence and enjoyment results. The plots depict the average participants' ratings on confidence (**a**) and enjoyment (**b**) across the 4 weeks of app training. Solid lines: mean; transparent regions: confidence interval. Light purple: healthy volunteers; dark purple: patients with obsessive-compulsive disorder (OCD). The insert plots show the results for the two subgroups of the OCD sample, when split based on their Yale-Brown Obsessive-Compulsive Scale (YBOCS) change after the app training (14 patients with improved symptomatology [reduced YBOCS scores] and 18 patients who remained stable or felt worse [i.e. respectively, unchanged or increased YBOCS scores]).

## Switch session

Similarly to the learning analysis of the practice sessions, we conducted an individual exponential fitting approach to the switch sessions and assessed between-group differences on performance, both on sequence duration (or movement time [$MT$]) and reaction time ($RT$, i.e. time taken to initiate the sequence, latency). No significant group differences were found in any of the three estimated fitting parameters for $MT$: *amount of learning ($MT_L$)*: $U$ = 395, $p$ = 0.082; *learning constant ($n_r$)*: $U$ = 575, $p$ = 0.544 and *asymptote ($MT_0$)*: $U$ = 450, $p$ = 0.311 (*Appendix 1—figure 2*, a). Descriptive statistics are the following (values provided as median and interquartile range): HV: $\tilde{MT_L}$ = 1.72 s, $IQR$ = 0.66 s; $\tilde{n_r}$ = 57, $IQR$ = 50; $\tilde{MT_0}$ = 1.56 s, $IQR$ = 0.41 s and OCD: $\tilde{MT_L}$ = 2.16 s, $IQR$ = 1.24 s; $\tilde{n_r}$ = 49, $IQR$ = 53; $\tilde{MT_0}$ = 1.67 s, $IQR$ = 0.45 s. Similarly, no significant group differences were found in the three estimated fitting parameters for $RT$ (*Appendix 1—figure 2b*): *reaction time speed-up* achieved over the course of the switch sessions ($R_L$): $U$ = 470, $p$ = 0.45; *reaction rate ($n_R$)*: $U$ = 450, $p$ = 0.31 and *reaction time at asymptote ($R_0$)*: $U$ = 552, $p$ = 0.75 (*Appendix 1—figure 2b*). Descriptive statistics are the following (values provided as median and interquartile range): HV: $\tilde{R_L}$ = 1.12 s, $IQR$ = 0.72 s; $\tilde{n_R}$ = 100, $IQR$ = 168; $\tilde{R_0}$ = 0.57 s, $IQR$ = 0.50 s and OCD: $\tilde{R_L}$ = 1.08 s, $IQR$ = 0.70 s; $\tilde{n_R}$ = 125, $IQR$ = 129; $\tilde{R_0}$ = 0.56 s, $IQR$ = 0.32 s. Since the switching between the two sequences was done in a pseudo-randomised order, we have also assessed $RT$ restricting our analysis to the specific trials where the switch occurred (i.e. when sequence 1 was followed by sequence 2 and vice versa). No group differences in the switching performance were found, meaning that patients with OCD could switch between the two sequences without any difficulties or slowness.

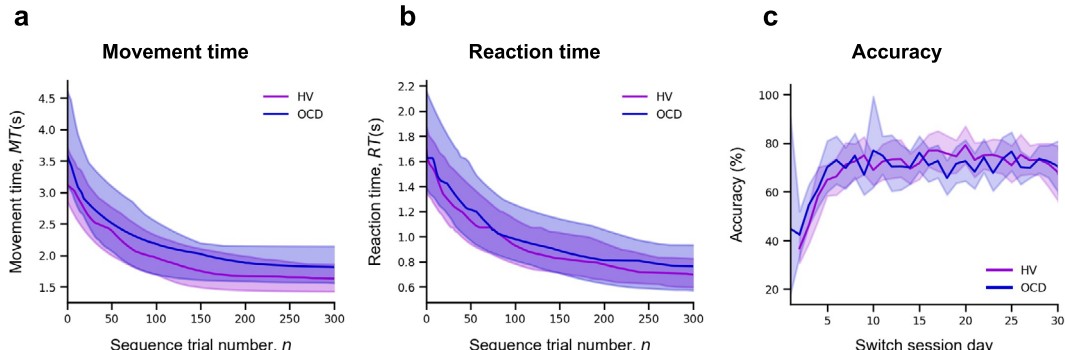

**Appendix 1—figure 2.** Model fitting procedure conducted for the switch sessions. Group comparison resulting from all individual exponential fits modeling the movement time (**a**), reaction time (**b**), and accuracy (**c**) profiles of each participant. No group differences were found. For (**a**) and (**b**) plots: solid lines represent median (*M*) and transparent regions the interquartile range (IQR); for plot (**c**): solid lines represent the mean and transparent regions for the confidence interval. Purple: healthy volunteers (HV); blue: patients with obsessive-compulsive disorder (OCD).

We also investigated potential group differences in accuracy during the switch sessions. After a similar individual exponential fitting approach, statistical analysis of the three estimated fitting parameters for accuracy indicated no group differences (*Appendix 1—figure 2c*): *amount of learning* as measured by accuracy achieved over the course of the switch sessions ($Acc_L$): $U$ = 538, $p$ = 0.56; *learning rate*: $U$ = 517, $p$ = 0.77 and *asymptote ($Acc_0$)*: $U$ = 419, $p$ = 0.29.

We did not analyze the short retention speed sessions because no new components were introduced here, as compared to the practice sessions. Therefore, we did not expect any differences between the practice and the speed sessions.

## Extinction

With the goal of promoting habitual actions, the app was designed to remove the explicit reward feedback (points) on the 21st day of practice (extinction procedure). We analyzed the effects of extinction on accuracy, sequence duration ($MT$), and reaction time ($RT$) by comparing the two blocks of practice pre- and post-removal of the rewarding feedback. We analyzed both the practice and switch conditions. While the latter had not previously received reward feedback, it could potentially have been influenced by extinction as a factor. A 2×2 ANOVA with extinction (pre- and post-extinction) as a within-subject factor and group as a between-subject factor indicated no group,

extinction, or interaction effects on accuracy, *RT*, or *MT* (*Appendix 1—figure 3*). Statistical results are as follows: (1) practice sessions; *variable reward*: no effect of group ($p$ = 0.31), extinction ($p$ = 0.28) or interaction ($p$ = 0.57) on accuracy; no effect of group ($p$ = 0.07), extinction ($p$ = 0.99) or interaction ($p$ = 0.61) on *MT*; no effect of group ($p$ = 0.06), extinction ($p$ = 0.53) or interaction ($p$ = 0.44) on reaction time; *continuous reward*: no effect of group ($p$ = 0.74), extinction ($p$ = 0.16) or interaction ($p$ = 0.82) on accuracy; no effect of group ($p$ = 0.56), extinction ($p$ = 0.78) or interaction ($p$ = 0.31) on *MT*; no effect of group ($p$ = 0.55), extinction ($p$ = 0.67) or interaction ($p$ = 0.47) on reaction time; (2) Switch sessions; no effect of group ($p$ = 0.19), extinction ($p$ = 0.17), or interaction ($p$ = 0.74) on accuracy; no effect of group ($p$ = 0.47), extinction ($p$ = 0.89), or interaction ($p$ = 0.27) on *MT*; no effect of group ($p$ = 0.46), extinction ($p$ = 0.78), or interaction ($p$ = 0.42) on reaction time.

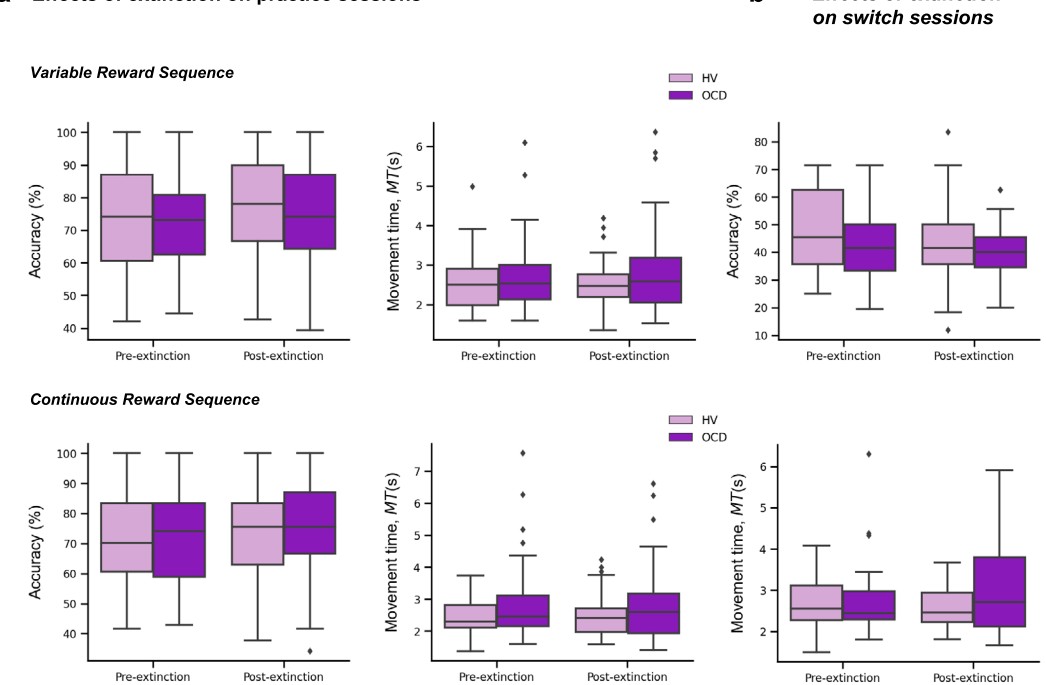

**Appendix 1—figure 3.** Extinction results. (**a**) Effects of extinction on the number of successful trials (accuracy) (left plots) and MT (middle plots) across the practice sessions only, separately for the variable (upper panel) and continuous (lower panel) reward conditions. (**b**) Effects of extinction on the number of successful trials (accuracy) (top plot) and MT (bottom plot) across the switch sessions. Note that for these analyses we used the two blocks of practice pre- and post-removal of the rewarding feedback (both on the practice and switch conditions).

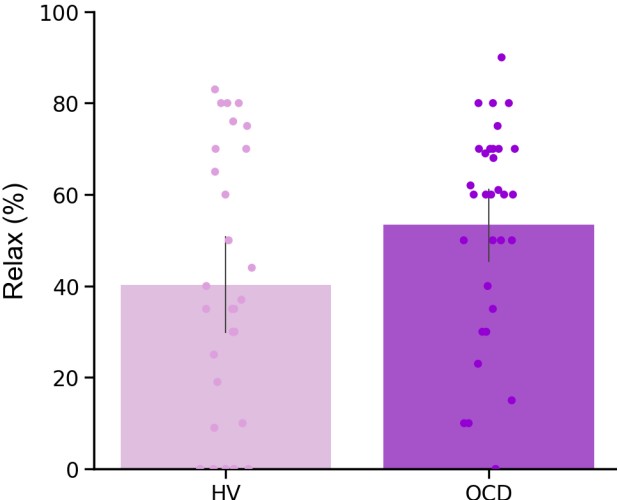

**Appendix 1—figure 4.** Relaxation ratings. Participant's ratings on how stressful/relaxing the app training was rated in a scale from –100% highly stressful to 100% very relaxing. No group differences ($U$ = 351, $p$ = 0.1).

## Sample size for the reward sensitivity analysis

The conditional probability distributions $p(\Delta T|\Delta R+)$ and $p(\Delta T|\Delta R-)$ were separately fitted to subsamples of the data across continuous reward practices (*Appendix 1—figure 5*). One practice corresponded to 20 correctly performed sequences. The $p(\Delta T|\Delta R)$ distributions for $MT$ (*Equation 7*) and the normalized consistency index $normC$ were fitted with a non-significantly different number of correct sequences in OCD and in the HV sample (mean 127 [SEM 12] sequences in OCD; 109 [9] sequences in HV; $BF$ < 1/3, moderate evidence against group differences in the number of correct sequences available for this analysis). However, more sequences were available to fit the $p(\Delta T|\Delta R+)$ distribution (mean 107 [SEM 10]) than the $p(\Delta T|\Delta R-)$ distribution (98 [8]). This outcome reflected that participants overall observed more increases than decrements in feedback scores, as expected. Importantly, matching both reward and group samples in the number of sequences yielded the same ANOVA results as reported in the main manuscript for the center and spread of the normalized $\Delta MT$ distribution and $normC$ distribution.

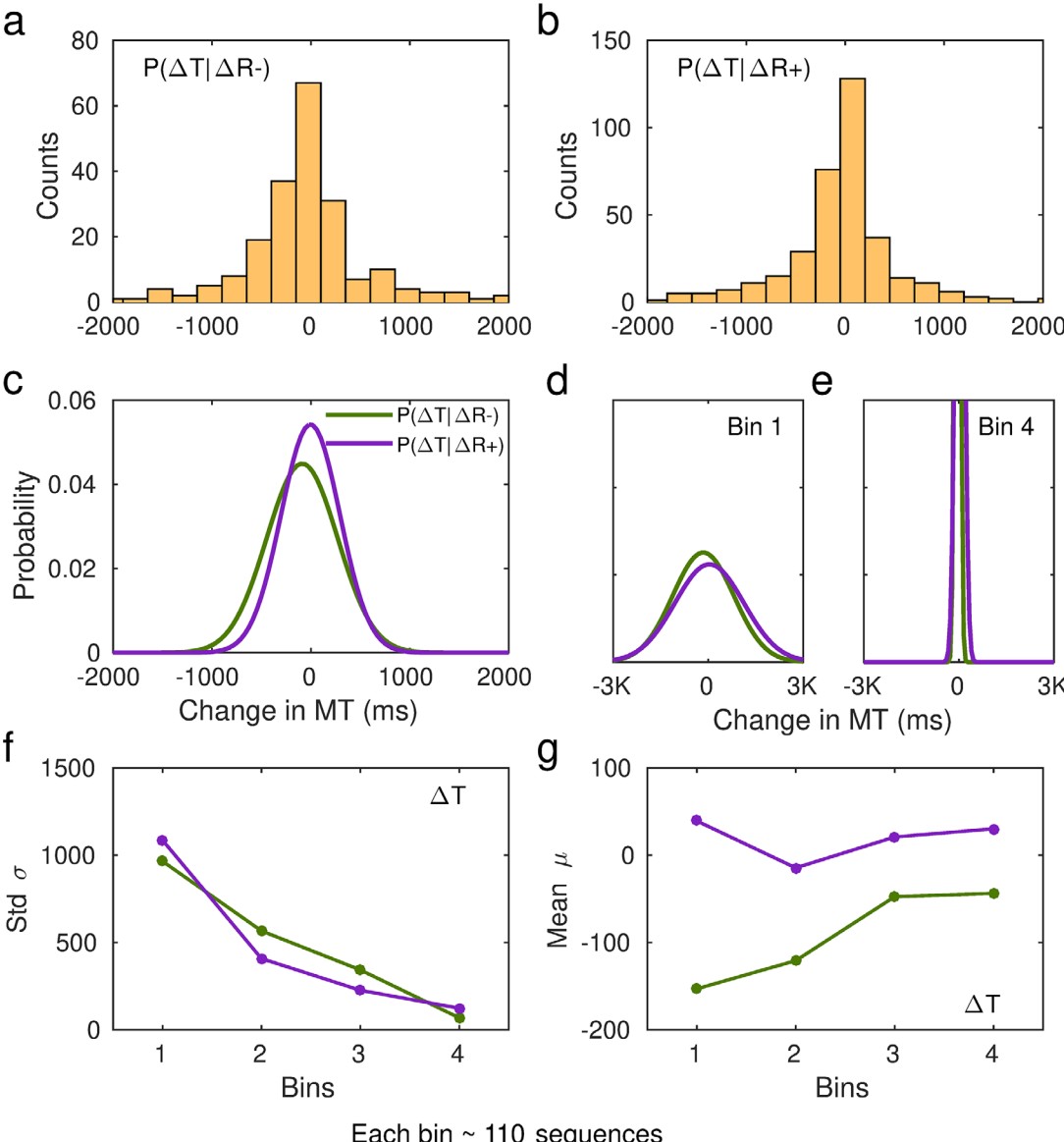

**Appendix 1—figure 5.** Fitting of performance data to conditional probability distributions p(ΔT|ΔR+) and p(ΔT|ΔR−). (**a**) Histogram of the changes in non-normalized movement time (*MT,* in ms, denoted by *T* in the graphic) following a decrease in scores relative to the previous trial, *ΔR−*. Data from one representative subject. (**b**) Same as (a) but for changes in *MT* following a reward increment, *ΔR+*. (**c**) Example of fitted Gaussian distributions to the histogram data in (**a**) and (**b**). The Gaussian fit was estimated in MATLAB using the Curve Fitting Toolbox [function fit, fit(x,y,'gauss1')] on a 20-bin histogram distribution of the data. This example illustrates the Gaussian fits to the total number of sequences and using non-normalized changes in *MT* (in ms). The full *p(ΔT|ΔR+)* and *p(ΔT|ΔR−)* distributions are denoted by the purple and green lines, respectively. (**d–e**) In our analyses, we split the total sample of correct sequences that each participant completed over the course of their training into four bins. We then fitted the Gaussian probability distributions to the subsample of sequences in each bin (~110 on average). The first bin is represented in (**d**), while the fourth bin is represented in (**e**). The y-axis limits are identical to those used in panel (**c**). (**f** and **g**) From the fitted Gaussian distributions we obtained the standard deviation, *σ*, to assess the spread of the distribution, and the mean, *μ*. Panel (**f**) displays the std across bins of sequences separately for *p(ΔT|ΔR+)* and *p(ΔT|ΔR−)*. Panel (**g**) shows the shift in the center of the Gaussian distribution over the course of practiced sequences.

Our analysis protocol was designed to ensure that incorrect trials do not contaminate or confound the results. To estimate the trial-to-trial difference in the normalized ΔMT (or *normC*) and ΔR, we exclusively included pairs of contiguous trials where participants achieved correct performance and

received feedback scores for both trials. For example, if a participant made a performance error on trial 23, we did not include ΔR or ΔMT estimates for the pairs of trials 23-22 and 24-23. Instead of excluding incorrect trials from our analyses, we retained them in our time series but assigned them an NaN (not a number) value in MATLAB. As a result, ΔR and ΔMT were not defined for those two pairs of trials. We followed the same protocol for *normC*. This approach ensured that our analyses are not confounded by incremental or decremental feedback scores between non-contiguous trials. In our previous work, when assessing the timing of correct actions during skilled sequence performance, we also considered events that were preceded and followed by correct actions. This excluded effects such as post-error slowing from contaminating our results (*Ruiz et al., 2009*; *Bury et al., 2019*).

**Appendix 1—table 1.** Participant's qualitative and quantitative feedback on the impact of the app-training in their life quality.

| Population | N | Qualitative feedback | Quantitative feedback (%) |
|---|---|---|---|
| HV | 28 | It was a very neutral game so it did not interfere positively or negatively with my life | 0 |
| | 2 | It did not impact my routines but it was very boring to do this every day | 0 |
| | 1 | I could see the progress and this was fulfilling | 10 |
| | 1 | It kept my brain active in the morning | 15 |
| | 1 | This was a month full of changes (I changed job, moved house, …) so the app was probably the only thing that remain constant throughout the month. I think it probably worked as some kind of mindfulness. | 60 |
| OCD | 16 | It was neutral, it did not interfere with my life in any way | 0 |
| | 4 | The app gave me a goal, some kind of structure to work towards. It was quite relaxing to do it. I enjoyed and repetition was not boring. | 2, 60, 70 and 70 |
| | 1 | It took my mind off the obsessions and rituals. Moreover, playing the app was a challenge to me. Seeing that I was getting better at it gave me a sense of achievement. This increased my confidence, which has spilled over into other areas of my life. | 80 |
| | 1 | I found it quite relaxing and a diversion at times. | 23 |
| | 1 | It was gratifying because I was finally able to complete something | 30 |
| | 1 | It kept me occupied | 10 |
| | 1 | The app made me use my brain and became part of my everyday routine | 60 |
| | 1 | It was definitely a stress relief. I was familiar enough with the app and this was very relaxing. I felt more confident throughout and then this makes me feel really good. I could switch off my obsessions. It was a focus although repetitive. The repetitiveness is relaxing because it is familiar. | 70 |
| | 1 | It took my mind off of other checking rituals for a short time. | 25 |
| | 1 | The app training made me sit and relax and because I had to concentrate I did not worry with other things. | 10 |
| | 1 | It was sometimes a bit of fun while having a difficult day | 30 |
| | 1 | While I was doing I was so focused on doing it that the thoughts were not coming so often. I just don't rate higher because in the last couple of weeks there were some other things in my life that upset me and the old came stronger | 50 |
| | 1 | It gave me sense of achievement | 40 |
| | 1 | Definitely not improved my life. A bit of the opposite as it was an extra thing I needed to do every day | −30 |

Quantitative feedback was given in a scale from −100% to 100%, in which 0 was no change in life quality, −100% was the maximum decrease in life quality and 100% was the maximum increase in life quality.

