## [Editor Report · eLife assessment]

This study provides **solid** evidence for differences in habit-learning in obsessive-compulsive disorder versus controls. Contrary to previous studies that employed a single laboratory session to study habit-learning, here a smartphone app delivered motor-sequence tasks daily for a month. These results have **important** implications for our understanding of goal-directed versus habit learning in obsessive-compulsive disorder.

---

## [Referee Report · Reviewer #1 (Public review)]

It is known that aberrant habit formation is a characteristic of obsessive-compulsive disorder (OCD). Habits can be defined according to the following features (Balleine and Dezfouli, 2019): rapid execution, invariant response topography, action 'chunking' and resistance to devaluation.

The extent to which OCD behavior is derived from enhanced habit formation relative to deficits in goal-directed behavior is a topic of debate in the current literature. This study examined habit-learning specifically (cf. deficits in goal-directed behavior) by regularly presenting, via smartphone, sequential learning tasks to patients with OCD and healthy controls. Participants engaged in the tasks every day over the course of a month. Automaticity, including the extent to which individual actions in the sequence become part of a unified 'chunk', was an important outcome variable. Following the 30 days of training, in-laboratory tasks were then administered to examine (1) if performing the learned sequences themselves had become rewarding (2) differences in goal-directed vs. habitual behavior.

Several hypotheses were tested, including:

Patients would have impaired procedural learning vs. healthy volunteers (this was not supported, possibly because there were fewer demands on memory in the task used here)

Once the task had been learned, patients would display automaticity faster (unexpectedly, patients were slower to display automaticity)

Habits would form faster under a continuous (vs. variable) reinforcement schedule

Exploratory analyses were also conducted: an interesting finding was that OCD patients with higher self-reported symptoms voluntarily completed more sessions with the habit-training app and reported a reduction in symptoms.

Strengths

This paper is well situated theoretically within the habit learning/OCD literature.

Daily training in a motor-learning task, delivered via smartphone, was innovative, ecologically valid and more likely to assay habitual behaviors specifically. Daily training is also more similar to studies with non-humans, making a better link with that literature. The use of a sequential-learning task (cf. tasks that require a single response) is also more ecologically valid.

The in-laboratory tests (after the 1 month of training) allowed the researchers to test if the OCD group preferred familiar, but more difficult, sequences over newer, simpler sequences.

Weaknesses

The authors were not able to test one criterion of habits, namely resistance to devaluation, due to the nature of the task.

The sample size was relatively small. Some potentially interesting individual differences within the OCD group could have been examined more thoroughly with a bigger sample (e.g., preference for familiar sequences). A larger sample may have allowed the statistical testing of any effects due to medication status.

The authors achieved their aims in that two groups of participants (patients with OCD and controls) engaged with the task over the course of 30 days. The repeated nature of the task meant that 'overtraining' was almost certainly established, and automaticity was demonstrated. This allowed the authors to test their hypotheses about habit learning. The results are supportive of the author's conclusions.

This article is likely to be impactful -- the delivery of a task across 30 days to a patient group is innovative and represents a new approach for the study of habit learning that is superior to an in-laboratory approach.

An interesting aspect of this manuscript is that it prompts a comparison with previous studies of goal-directed/habitual responding in OCD that used devaluation protocols, and which may have had their effects due to deficits in goal-directed behavior and not enhanced habit learning per se.

---

## [Author Response]

The following is the authors’ response to the previous reviews.

**Public Reviews:**

**Reviewer #2 (Public Review):**
I would like to express my appreciation for the authors' dedication to revising the manuscript. It is evident that they have thoughtfully addressed numerous concerns I previously raised, significantly contributing to the overall improvement of the manuscript.

Response: We appreciate the reviewers’ recognition of our efforts in revising the manuscript.

My primary concern regarding the authors' framing of their findings within the realm of habitual and goal-directed action control persists. I will try explain my point of view and perhaps clarify my concerns. While acknowledging the historical tendency to equate procedural learning with habits, I believe a consensus has gradually emerged among scientists, recognizing a meaningful distinction between habits and skills or procedural learning. I think this distinction is crucial for a comprehensive understanding of human action control. While these constructs share similarities, they should not be used interchangeably. Procedural learning and motor skills can manifest either through intentional and planned actions (i.e., goal-directed) or autonomously and involuntarily (habitual responses).

Response: We would like to clarify that, contrary to the reviewer’s assertion of a scientific consensus on this matter, the discussion surrounding the similarities and differences between habits and skills remains an ongoing and unresolved topic of interest among scientists (Balleine and Dezfouli, 2019; Du and Haith, 2023; Graybiel and Grafton, 2015; Haith and Krakauer, 2018; Hardwick et al., 2019; Kruglanski and Szumowska, 2020; Robbins and Costa, 2017). We absolutely agree with the reviewer that “Procedural learning and motor skills can manifest either through intentional and planned actions (i.e., goal-directed) or autonomously and involuntarily (habitual responses)”. But so do habits. Some researchers also highlight the intentional/goal-directed nature of habits e.g., Du and Haith, 2023, “Habits are not automatic” (preprint) or Kruglanski and Szumowska, 2020, “Habitual behavior is goal-driven”: “definitions of habits that include goal independence as a foundational attribute of habits are begging the question; they effectively define away, and hence dispose of, the issue of whether habits are goal-driven (p 1258).” Therefore, there is no clear consensus concerning the concept of habit.

While we acknowledge the meaningful distinctions between habits and skills, we also recognize a substantial body of literature supporting the overlap between these concepts (cited in our manuscript), particularly at the neural level. The literature clearly indicates that both habits and skills are mediated by subcortical circuits, with a progressive disengagement of cognitive control hubs in frontal and cingulate cortices as repetition evolves. We do not use these concepts interchangeably. Instead, we simply present evidence supporting the assertion that our trained app sequences meet several criteria for their habitual nature.

Our choice of Balleine and Dezfouli (2018)'s criteria stemmed from the comprehensive nature of their definitions, which effectively synthesized insights from various researchers (Mazar and Wood, 2018; Verplanken et al., 1998; Wood, 2017, etc). Importantly, their list highlights the positive features of habits that were previously overlooked. However, these authors still included a controversial criterion ("habits as insensitive to changes in their relationship to their individual consequences and the value of those consequences"), even though they acknowledged the problems of using outcome devaluation methods and of relying on a null-effect. According to Kruglanski andSzumowska (2020), this criterion is highly problematic as “If, by definition, habits are goalindependent, then any behavior found to be goal-dependent could not be a habit on sheer logical grounds” (p. 1257). In their definition, “habitual behavior is sensitive to the value of the reward (i.e., the goal) it is expected to mediate and is sensitive to the expectancy of goal attainment (i.e., obtainment of the reward via the behavior, p.1265). In fact, some recent analyses of habitual behavior are not using devaluation or revaluation as a criterion (Du and Haith, 2023). This article, for example, ascertains habits using different criteria and provides supporting evidence for trained action sequences being understood as skills, with both goal-directed and habitual components.

In the discussion of our manuscript, we explicitly acknowledge that the app sequences can be considered habitual or goal-directed in nature and that this terminology does not alter the fact that our overtrained sequences exhibit clear habitual features.

Watson et al. (2022) aptly detailed my concerns in the following statements: "Defining habits as fluid and quickly deployed movement sequences overlaps with definitions of skills and procedural learning, which are seen by associative learning theorists as different behaviors and fields of research, distinct from habits.""...the risk of calling any fluid behavioral repertoire 'habit' is that clarity on what exactly is under investigation and what associative structure underpins the behavior may be lost."I strongly encourage the authors, at the very least, to consider Watson et al.'s (2022) suggestion: "Clearer terminology as to the type of habit under investigation may be required by researchers to ensure that others can assess at a glance what exactly is under investigation (e.g., devaluationinsensitive habits vs. procedural habits)", and to refine their terminology accordingly (to make this distinction clear). I believe adopting clearer terminology in these respects would enhance the positioning of this work within the relevant knowledge landscape and facilitate future investigations in the field.

Response: We would like to highlight that we have indeed followed Watson et al (2022)’s recommendations on focusing on other features/criteria of habits at the expense of the outcome devaluation/contingency degradation paradigm, which has been more controversial in the human literature. Our manuscript clearly aligns with Watson et al. (2022) ‘s recommendations: “there are many other features of habits that are not captured by the key metrics from outcome devaluation/contingency degradation paradigms such as the speed at which actions are performed and the refined and invariant characteristics of movement sequences (Balleine and Dezfouli, 2019). Attempts are being made to develop novel behavioral tasks that tap into these positive features of habits, and this should be encouraged as should be tasks that are not designed to assess whether that behavior is sensitive to outcome devaluation, but capture the definition of habits through other measures”.

Regarding the authors' use of Balleine and Dezfouli's (2018) criteria to frame recorded behavior as habitual, as well as to acknowledgment the study's limitations, it's important to highlight that while the authors labelled the fourth criterion (which they were not fulfilling) as "resistance to devaluation," Balleine and Dezfouli (2018) define it as "insensitive to changes in their relationship to their individual consequences and the value of those consequences." In my understanding, this definition is potentially aligned with the authors' re-evaluation test, namely, it is conceptually adequate for evaluating the fourth criterion (which is the most accepted in the field and probably the one that differentiate habits from skills). Notably, during this test, participants exhibited goaldirected behavior.The authors characterized this test as possibly assessing arbitration between goal-directed and habitual behavior, stating that participants in both groups "demonstrated the ability to arbitrate between prior automatic actions and new goal-directed ones." In my perspective, there is no justification for calling it a test of arbitration. Notably, the authors inferred that participants were habitual before the test based on some criteria, but then transitioned to goal-directed behavior based on a different criterion. While I agree with the authors' comment that: "Whether the initiation of the trained motor sequences in experiment 3 (arbitration) is underpinned by an action-outcome association (or not) has no bearing on whether those sequences were under stimulus-response control after training (experiment 1)." they implicitly assert a shift from habit to goal-directed behavior without providing evidence that relies on the same probed mechanism.Therefore, I think it would be more cautious to refer to this test as solely an outcome revaluation test. Again, the results of this test, if anything, provide evidence that the fourth criterion was tested but not met, suggesting participants have not become habitual (or at least undermines this option).

Response: In our previously revised manuscript, we duly acknowledged that the conventional (perhaps nowadays considered outdated) goal devaluation criterion was not met, primarily due to constraints in designing the second part of the study. We did cite evidence from another similar study that had used devaluation app-trained action sequences to demonstrate habitual qualities (but the reviewer ignored this).

The reviewer points out that we did use a manipulation of goal revaluation in one of the follow-up tests conducted (although this was not a conventional goal revaluation test inasmuch that it was conducted in a novel context). In this test, please note that we used 2 manipulations: monetary and physical effort. Although we did show that subjects, including OCD patients, were apparently goaldirected in the monetary reward manipulation, this was not so clear when goal re-evaluation involved the physical effort expended. In this effort manipulation, participants were less goaloriented and OCD patients preferred to perform the longer, familiar, to the shorter, novel sequence, thus exhibiting significantly greater habitual tendencies, as compared to controls. Hence, we cannot decisively conclude that the action sequence is goal-directed as the reviewer is arguing. In fact, the evidence is equivocal and may reflect both habitual and goal-directed qualities in the performance of this sequence, consistent with recent interpretations of skilled/habitual sequences (Du and Haith, 2023). Relying solely on this partially met criterion to conclude that the app-trained sequences are goal-directed, and therefore not habitual, would be an inaccurate assessment for several reasons: (1) the action sequences did satisfy all other criteria for being habitual; (2) this approach would rest on a problematic foundation for defining habits, as emphasized by Kruglanski & Szumowska (2020); and (3) it would succumb to the pitfall of subscribing to a zero-sum game perspective, as cautioned by various researchers, including the review by Watson et al. (2022) cited by the referee, thus oversimplifying the nuanced nature of human behavior.

While we have previously complied with the reviewer’s suggestion on relabelling our follow-up test as a “revaluation test” instead of an “arbitration test”, we have now explicitly removed all mentions of the term “arbitration” (which seems to raise concerns) throughout the manuscript. As the reviewer has suggested, we now use a more refined terminology by explicitly referring to the measured behavior as "procedural habits", as he/she suggested. We have also extensively revised the discussion section of our manuscript to incorporate the reviewer’s viewpoint. We hope that these adjustments enhance the clarity and accuracy of our manuscript, addressing the concerns raised during this review process.

In essence, this is an ontological and semantic matter, that does not alter our findings in any way.Whether the sequences are consider habitual or goal directed, does not change our findings that (1) Both groups displayed equivalent procedural learning and automaticity attainment; (2) OCD patients exhibit greater subjective habitual tendencies via self-reported questionnaires; (3) Patients who had elevated compulsivity and habitual self-reported tendencies engaged significantly more with the motor habit-training app, practiced more and reported symptom relief at the end of the study; (4) these particular patients also show an augmented inclination to attribute higher intrinsic value to familiar actions, a possible mechanism underlying compulsions.

**Reviewer #2 (Recommendations For The Authors):**
A few more small comments (with reference to the point numbers indicated in the rebuttal):(14) I am not entirely sure why the suggested analysis is deemed impractical (i.e., why it cannot be performed by "pretending" participants received the points they should have received according to their performance). This can further support (or undermine) the idea of effect of reward on performance rather than just performance on performance.

Response: We have now conducted this analysis, generating scores for each trial of practices after day 20, when participants no longer gained points for their performance. This analysis assesses whether participants trial-wise behavioral changes exhibit a similar pattern following simulated relative increases or decrease in scores, as if they had been receiving points at this stage. Note that this analysis has fewer trials available, around 50% less on average.

Before presenting our results, we wish to emphasize the importance of distinguishing between the effects of performance on performance and the effects of reward on performance. In response to a reviewer's suggestion, we assessed the former in the first revision of our manuscript. We normalized the movement time variable and evaluated how normalized behavioral changes responded to score increments and decrements. The results from the original analyses were consistent with those from the normalized data.

Regarding the phase where participants no longer received scores, we believe this phase primarily helps us understand the impact of 'predicted' or 'learned' rewards on performance. Once participants have learned the simple association between faster performance and larger scores, they can be expected to continue exhibiting the reward sensitivity effects described in our main analysis. We consider it is not feasible to assess the effects of performance on performance during the reward removal phase, which occurs after 20 days. Therefore, the following results pertain to how the learned associations between faster movement times and scores persist in influencing behavior, even when explicit scores are no longer displayed on the screen.

Results: The main results of the effect of reward on behavioral changes persist, supporting that relative increases or decreases in scores (real or imagined/inferred) modulate behavioral adaptations trial-by-trial in a consistent manner across both cohorts. The direction of the effects of reward is the same as in the main analyses presented in the manuscript: larger mean behavioral changes (smaller std) following ∆R- .First, concerning changes in “normalized” movement time (MT) trial-by-trial, we conducted a 2 x 2 factorial analysis of the centroid of the Gaussian distributions with the same factors Reward, Group and Bin. This analysis demonstrated a significant main effect of Reward (P = 2e-16), but not of Group (P = 0.974) or Bin (P = 0.281). There were no significant interactions between factors. The main Reward effect can be observed in the top panel of the figure below.The same analysis applied to the spread (std) of the Gaussian distributions revealed a significant main effect of Reward (P = 0.000213), with no additional main effects or interactions.

**Author response image 1. sa2fig1:** 

Next, conducting the same 2 x 2 factorial analyses on the centroid and spread of the Gaussian distributions fitted to the Consistency data, we also obtained a robust significant main effect of Reward. For the centroid variable, we obtained a significant main effect of Reward (P = 0.0109) and Group (P = 0.0294), while Bin and the factor interactions were non-significant. See the top panel of the figure below.

On the other hand, Reward also modulated significantly the spread of the Gaussian distributions fitted to the Consistency data, P = 0.00498. There were no additional significant main effects or interactions. See the bottom panel in the figure below.

Note that here the factorial analysis was performed on the logarithmic transformation of the std.

(16) I find this result interesting and I think it might be worthwhile to include it in the paper.

Response: We have now included this result in our revised manuscript (page 28)

(18) I referred to this sentence: "The app preferred sequence was their preferred putative habitual sequence while the 'any 6' or 'any 3'-move sequences were the goal-seeking sequences." In my understanding, this implies one choice is habitual and another indicates goal-directedness.One last small comment:In the Discussion it is stated: "Moreover, when faced with a choice between the familiar and a new, less effort-demanding sequence, the OCD group leaned toward the former, likely due to its inherent value. These insights align with the theory of goal-direction/habit imbalance in OCD (Gillan et al., 2016), underscoring the dominance of habits in particular settings where they might hold intrinsic value."This could equally be interpreted as goal-directed behavior, so I do not think there is conclusive support for this claim.

Response: The choice of the familiar/trained sequence, as opposed to the 'any 6' or 'any 3'-move sequences cannot be explicitly considered goal-directed: firstly, because the app familiar sequences were associated with less monetary reward (in the any-6 condition), and secondly, because participants would clearly need more effort and time to perform them. Even though these were automatic, it would still be much easier and faster to simply tap one finger sequentially 6 times (any 6) or 3 times (any-3). Therefore, the choice for the app-sequence would not be optimal/goaldirected. In this sense, that choice aligns with the current theory of goal-direction/habit imbalance of OCD. We found that OCD patients prefer to perform the trained app sequences in the physical effort manipulation (any-3 condition). While this, on one hand cannot be explicitly considered a goal-directed choice, we agree that there is another possible goal involved here, which links to the intrinsic value associated to the familiar sequence. In this sense the action could potentially be considered goal-directed. This highlights the difficulty of this concept of value and agrees with: (1) Hommel and Wiers (2017): “Human behavior is commonly not driven by one but by many overlapping motives . . . and actions are commonly embedded into larger-scale activities with multiple goals defined at different levels. As a consequence, even successful satiation of one goal or motive is unlikely to also eliminate all the others(p. 942) and (2) Kruglanski & Szumowska (2020)’s account that “habits that may be unwanted from the perspective of an outsider and hence “irrational” or purposeless, may be highly wanted from the perspective of the individual for whom a habit is functional in achieving some goal” (p. 1262) and therefore habits are goal-driven.

References:

Balleine BW, Dezfouli A. 2019. Hierarchical Action Control: Adaptive Collaboration Between Actions and Habits. Front Psychol 10:2735. doi:10.3389/fpsyg.2019.02735

Du Y, Haith A. 2023. Habits are not automatic. doi:10.31234/osf.io/gncsf Graybiel AM, Grafton ST. 2015. The Striatum: Where Skills and Habits Meet. Cold Spring Harb Perspect Biol 7:a021691. doi:10.1101/cshperspect.a021691

Haith AM, Krakauer JW. 2018. The multiple effects of practice: skill, habit and reduced cognitive load. Current Opinion in Behavioral Sciences 20:196–201. doi:10.1016/j.cobeha.2018.01.015

Hardwick RM, Forrence AD, Krakauer JW, Haith AM. 2019. Time-dependent competition between goal-directed and habitual response preparation. Nat Hum Behav 1–11. doi:10.1038/s41562019-0725-0

Hommel B, Wiers RW. 2017. Towards a Unitary Approach to Human Action Control. Trends Cogn Sci 21:940–949. doi:10.1016/j.tics.2017.09.009

Kruglanski AW, Szumowska E. 2020. Habitual Behavior Is Goal-Driven. Perspect Psychol Sci 15:1256– 1271. doi:10.1177/1745691620917676

Mazar A, Wood W. 2018. Defining Habit in Psychology In: Verplanken B, editor. The Psychology of Habit: Theory, Mechanisms, Change, and Contexts. Cham: Springer International Publishing. pp. 13–29. doi:10.1007/978-3-319-97529-0_2

Robbins TW, Costa RM. 2017. Habits. Current Biology 27:R1200–R1206. doi:10.1016/j.cub.2017.09.060

Verplanken B, Aarts H, van Knippenberg A, Moonen A. 1998. Habit versus planned behaviour: a field experiment. Br J Soc Psychol 37 ( Pt 1):111–128. doi:10.1111/j.2044-8309.1998.tb01160.x

Watson P, O’Callaghan C, Perkes I, Bradfield L, Turner K. 2022. Making habits measurable beyond what they are not: A focus on associative dual-process models. Neurosci Biobehav Rev 142:104869. doi:10.1016/j.neubiorev.2022.104869

Wood W. 2017. Habit in Personality and Social Psychology. Pers Soc Psychol Rev 21:389–403. doi:10.1177/1088868317720362